# Gut microbiota carcinogen metabolism causes distal tissue tumours

Blanka Roje[1,9], Boyao Zhang[2,9], Eleonora Mastrorilli[2], Ana Kovačić[3], Lana Sušak[1], Ivica Ljubenkov[4], Elena Ćosić[1], Katarina Vilović[5], Antonio Meštrović[6], Emilija Lozo Vukovac[7], Viljemka Bučević-Popović[4], Željko Puljiz[6], Ivana Karaman[5], Janoš Terzić[1✉] & Michael Zimmermann[2,8✉]

Exposure to environmental pollutants and human microbiome composition are important predisposition factors for tumour development[1,2]. Similar to drug molecules, pollutants are typically metabolized in the body, which can change their carcinogenic potential and affect tissue distribution through altered toxicokinetics[3]. Although recent studies demonstrated that human-associated microorganisms can chemically convert a wide range of xenobiotics and influence the profile and tissue exposure of resulting metabolites[4,5], the effect of microbial biotransformation on chemical-induced tumour development remains unclear. Here we show that the depletion of the gut microbiota affects the toxicokinetics of nitrosamines, which markedly reduces the development and severity of nitrosamine-induced urinary bladder cancer in mice[6,7]. We causally linked this carcinogen biotransformation to specific gut bacterial isolates in vitro and in vivo using individualized bacterial culture collections and gnotobiotic mouse models, respectively. We tested gut communities from different human donors to demonstrate that microbial carcinogen metabolism varies between individuals and we showed that this metabolic activity applies to structurally related nitrosamine carcinogens. Altogether, these results indicate that gut microbiota carcinogen metabolism may be a contributing factor for chemical-induced carcinogenesis, which could open avenues to target the microbiome for improved predisposition risk assessment and prevention of cancer.

Increasing evidence suggests that human-associated microorganisms, collectively called the human microbiota, play an important role in cancer development and progression[8,9]. Although the molecular mechanisms of microbial contributions to cancer development remain largely unknown, translocation of tumour-inducing bacteria from the gut into tissues and microbial production of genotoxins (that is, colibactin) have been demonstrated to induce cancer development[10,11]. Another important carcinogenic mechanism is the exposure to environmental pollutants, which are often further metabolized after entering the human body[3]. Contaminant-metabolizing human enzymes, such as cytochrome P450 enzymes, have been well described for their capacity to convert a broad variety of carcinogens, which can lead to their activation or inactivation[12]. Recent studies have demonstrated that human gut bacteria also have a large potential to metabolize chemically diverse compounds, such as medical drugs, and that the resulting products can get distributed to distal tissues[4,5]. This led us to investigate whether the gut microbiota could contribute to tumour onset and development through the metabolism of environmental carcinogens.

We exemplarily studied bladder cancer, the tenth most common cancer worldwide with high recurrence and progression rates in its invasive form[13], whose major risk factors are environmental and occupational exposure to carcinogens[14]. Nitrosamine compounds, such as *N*-butyl-*N*-(4-hydroxybutyl)-nitrosamine (BBN) and related organic contaminants in tobacco smoke, have been well described to induce bladder cancer[6]. In fact, chronic exposure to BBN is routinely used to induce bladder cancer in rodent models[7]. Following oral administration, BBN undergoes substantial metabolism through oxidation and glucuronide conjugation. Notably, *N*-butyl-*N*-(3-carboxypropyl)-nitrosamine (BCPN), the oxidation product of BBN, induces tumorigenesis through DNA adduct formation in the urothelium[6,15]. We utilized the BBN-induced bladder cancer mouse model to investigate the contribution of the gut microbiota to chemically induced tumorigenesis as well as its role in toxicokinetics (Fig. 1a).

## Antibiotics reduce bladder tumours

To test for possible gut microbiota contributions to BBN-induced bladder cancer development, we compared bladder tumour pathology of antibiotic (ABX)-treated and non-treated C57BL/6 mice with concomitant BBN administration through drinking water (0.05% v/v).

[1]Laboratory for Cancer Research, University of Split School of Medicine, Split, Croatia. [2]Structural and Computational Biology Unit, European Molecular Biology Laboratory, Heidelberg, Germany. [3]Public Health Institute of Split and Dalmatia County, Split, Croatia. [4]Department of Chemistry, University of Split Faculty of Science, Split, Croatia. [5]Department of Pathology, University Hospital of Split, Split, Croatia. [6]Department of Gastroenterology, University Hospital of Split, Split, Croatia. [7]Department of Pulmonology, University Hospital of Split, Split, Croatia. [8]Molecular Medicine Partnership Unit (MMPU), University of Heidelberg and European Molecular Biology Laboratory (EMBL), Heidelberg, Germany. [9]These authors contributed equally: Blanka Roje, Boyao Zhang. ✉e-mail: janos.terzic@mefst.hr; michael.zimmermann@embl.de

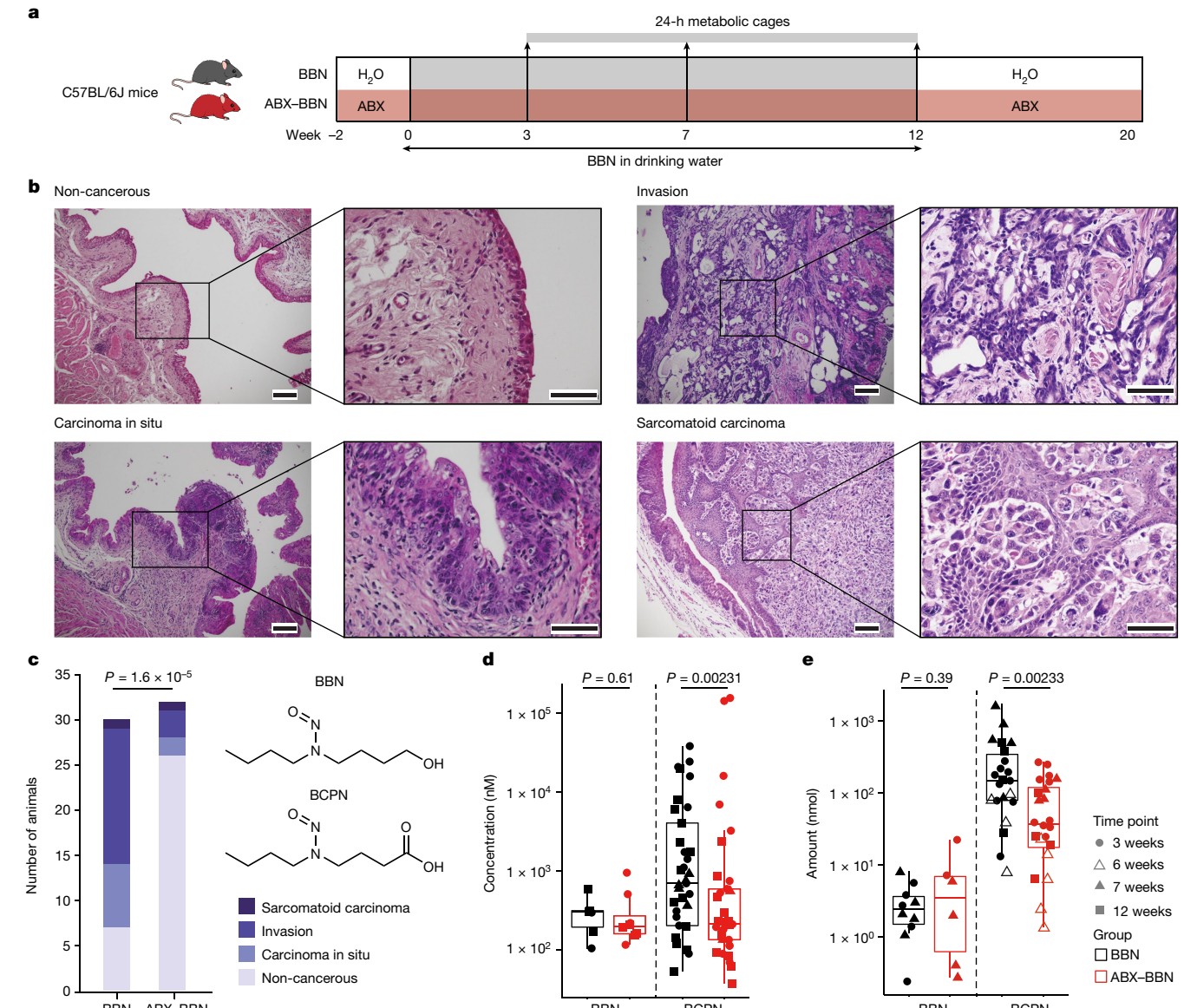

**Fig. 1 | ABX treatment reduces chemical carcinogenesis in mice. a**, Design of the mouse experiment to study BBN-induced bladder cancer. **b**, Representative bladder histology after 20 weeks of BBN administration in drinking water used to classify tumour stages in situ. The experiment was repeated independently five times with comparable results (number of animals at each bladder cancer development stage is available in Supplementary Table 3 and all histology images are shown in Extended Data Fig. 1). Scale bars, for each pair of images, 100 μm (left) and 50 μm (right). **c**, Left: quantification of animals with a given bladder cancer developmental stage in respective groups ($n$ = 32 and 30, two-sided $\chi^2$ test with Yates continuity correction; Supplementary Table 3). Right: chemical structures of BBN and BCPN. **d**, BBN and BCPN concentration in bladder tissue after 3, 7 and 12 weeks of BBN treatment without ($n$ = 37) or with ($n$ = 38) ABX treatment; $P$ values were calculated with two-sided Wilcoxon test (Supplementary Table 4, measurements for BBN in one of the 12-week batches were not obtained, and hence for BBN, $n$ = 33 and 33; owing to log scale, data points that are at 0 are not shown). **e**, BBN and BCPN accumulation in urine collected in metabolic cages over 24 h at time points 3, 6, 7 and 12 weeks of BBN treatment ($n$ = 22 and 24); $P$ values were calculated with two-sided Wilcoxon test (Supplementary Table 5, measurements for BBN in the 12-week batch were not obtained, and hence for BBN, $n$ = 10 and 9). For all box plots, the lower and upper hinges depict the 25th and 75th percentiles. The upper and lower whiskers extend from the hinge to the largest and smallest values at most 1.5 × interquartile range (IQR) from the hinge. The centre line in the box shows the median.

We confirmed a reduction of the bacterial load in the gut following ABX treatment through colony-forming unit counting, which was 99.99% lower in treated versus non-treated animals (Extended Data Fig. 1a and Supplementary Table 1). Additionally, 16S rRNA amplicon sequencing demonstrated that ABX treatment, but not BBN administration, significantly reduced gut microbiome α-diversity (measured as observed number of species: Kruskal–Wallis test, $P$ value 0.008134; or as Shannon diversity: Kruskal–Wallis test, $P$ value 4.86 × 10$^{-5}$), and significantly altered the overall community composition (permutational multivariate analysis of variance test statistic 699.172, $P$ value

0.001, 999 permutations; Extended Data Fig. 1b–d and Supplementary Table 2). After 12 weeks of continuous exposure to BBN in drinking water, BBN was removed for 8 weeks and blinded pathological analysis of urinary bladders was carried out (Fig. 1b and Supplementary Data 1). Notably, most mice with reduced gut bacterial load (ABX–BBN group) were completely free of bladder tumour pathology (81%, $n$ = 26/32), whereas 77% of the control mice without ABX treatment (BBN group) developed neoplastic changes in their bladder tissue ($n$ = 23/30, $\chi^2$ test, $P$ value = 1.6 × 10$^{-5}$; Fig. 1c and Supplementary Table 3). Moreover, invasive tumours, the most aggressive tumour presentation, were found in

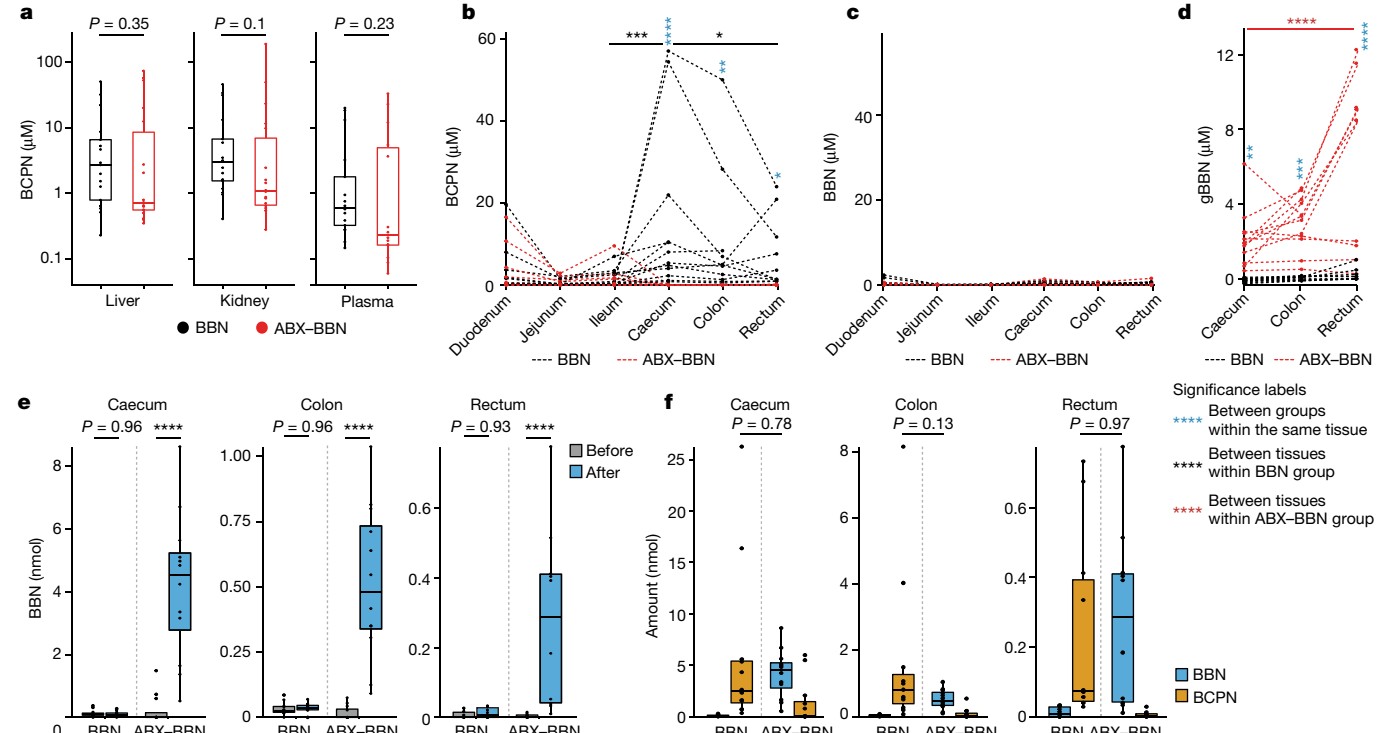

**Fig. 2 | BCPN is generated in the large intestine from BBN and its derivatives.**
**a**, BCPN concentration in liver, kidney and plasma in mice treated with ABX–BBN (*n* = 18; red) and BBN (*n* = 17; black; two-sided Wilcoxon test; Supplementary Table 6). **b–d**, BCPN (**b**), BBN (**c**) and gBBN (**d**) levels along the gastrointestinal tract of ABX–BBN (red, *n* = 12) and BBN (black, *n* = 12) groups (Supplementary Table 7). Pairwise comparisons between groups for the same tissue and between tissues in the same group were carried out with estimated marginal means (EMMs) after fitting the data with a linear mixed model, and *P* values were calculated with a two-sided *t*-test and adjusted for multiple testing using the false discovery rate (FDR; Supplementary Table 9). **e**, Quantification of BBN in caecum, colon and rectum before and after in vitro β-glucuronidase treatment (Supplementary Table 8). Comparisons between groups were carried out with

EMMs after fitting the data with a linear mixed model, and *P* values were calculated with a two-sided *t*-test and adjusted for multiple testing with FDR (Supplementary Table 10). **f**, Mass balancing between BBN and BCPN levels in caecum, colon and rectum following in vitro β-glucuronidase treatment (Supplementary Table 8). Pairwise comparisons between BBN and BCPN in different groups were carried out with EMMs and a two-sided Tukey test (Supplementary Table 11). Results are shown for mice treated with BBN for 3 weeks. *$P \leq 0.05$, **$P \leq 0.01$, ***$P \leq 0.001$, ****$P \leq 0.0001$. For all box plots, the lower and upper hinges depict the 25th and 75th percentiles. The upper and lower whiskers extend from the hinge to the largest and smallest values at most 1.5 × IQR from the hinge. The centre line in the box shows the median.

53% (*n* = 16/30, $\chi^2$ test, *P* value = $1.5 \times 10^{-3}$) of mice in the BBN group, but only in 12% (*n* = 4/32) of the ABX–BBN group. These results indicated that the gut microbiota may enhance bladder cancer development in BBN-exposed mice.

Next we examined whether contributions of the gut microbiota to bladder cancer development lie in altered urothelial tissue exposure to the administered carcinogen. Using liquid chromatography-coupled mass spectrometry quantification, we found that concentrations of BCPN, but not BBN, in bladder tissue of ABX–BBN-treated mice were significantly lower (*P* value = $2.31 \times 10^{-3}$) than in BBN-treated mice during the 12 weeks of BBN exposure (Fig. 1d and Supplementary Table 4). Additionally, we quantified urinary elimination of BBN and BCPN in metabolic cages over 24 h. These measurements demonstrated that urinary BCPN elimination was significantly (*P* value = $2.33 \times 10^{-3}$) lower in the ABX–BBN compared to BBN mice, whereas BBN elimination was comparable between the two mouse groups (Fig. 1e and Supplementary Table 5). These results link the observed decrease of tumorigenesis following reduction of the gut bacterial load to decreased urinary BCPN elimination and urothelial BCPN exposure.

## BCPN is produced in the intestinal tract

To elucidate possible links between the gut microbiota and BCPN production, we quantified BCPN in nine additional body compartments (plasma, liver, kidney, duodenum, jejunum, ileum, caecum,

colon and rectum) of BBN and ABX–BBN-treated animals after 3 and 7 weeks of BBN exposure (Fig. 1a). Whereas we did not detect any difference in BCPN concentrations in plasma, liver and kidney between the two groups of mice (Fig. 2a, Extended Data Fig. 2a and Supplementary Tables 6 and 12), BCPN accumulated in caecum and colon luminal contents of BBN-treated, but not ABX–BBN-treated, animals (Fig. 2b, Extended Data Fig. 2b and Supplementary Tables 7, 9, 13 and 14). We also observed that BCPN levels decrease from caecum to rectum, suggesting BCPN reabsorption from the lower intestine. To control for the effect of ABX on host BBN metabolism, we treated germ-free mice with BBN in drinking water for 10 days with or without ABX treatment and compared BCPN levels in bladder, liver, kidney, plasma and caecum. BCPN levels in all of these tissues were comparable with or without ABX treatment (Extended Data Fig. 2d and Supplementary Table 15) suggesting that BBN metabolism and toxicokinetics is not affected by ABX. To directly test for potential effects of ABX on host BBN metabolism, we measured hepatic BBN to BCPN conversion in HEP-G2 cells, and found it to be unaffected by the addition of ABX (Extended Data Fig. 2e and Supplementary Table 16). These control experiments consolidated our hypothesis that the difference in BCPN levels between the BBN- and ABX–BBN-treated animals is not caused by ABX, but that BCPN is produced in a microbiome-dependent manner in the lower intestine.

Next we investigated whether the observed BCPN production in caecum results from intestinal BBN oxidation. In fact, BBN concentrations in the lower intestine were more than a magnitude lower in ABX-treated

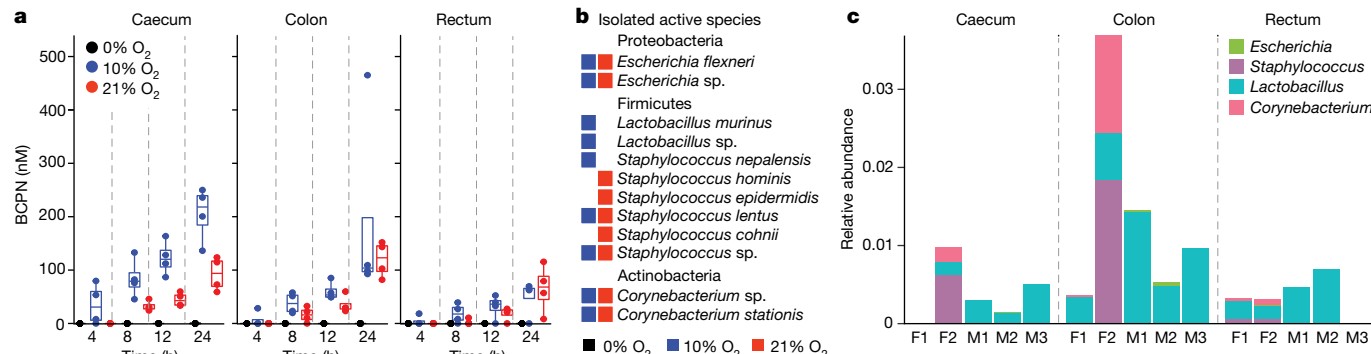

**Fig. 3 | Ex vivo BBN-to-BCPN conversion by gut bacterial communities and isolates from mice. a**, BCPN production over 24 h of incubation with BBN in ex vivo community cultures of luminal content of mouse caecum, colon and rectum under anaerobic (0% oxygen), microaerobic (10% oxygen) and aerobic (21% oxygen) conditions (Supplementary Table 19). Each data point represents the average of four technical replicates from each of four mice, two of each sex.

**b**, Bacterial isolates capable of BBN-to-BCPN conversion that could be mapped to the same microbiomes that were subjected to 16S rRNA sequencing (Supplementary Table 21). **c**, Relative abundances of the active genera from **b** in the respective microbiomes from different mice (F, female; M, male; Supplementary Table 22).

animals (Fig. 2c, Extended Data Fig. 2c and Supplementary Tables 7 and 13), suggesting that free BBN could account for only a small fraction of the intestinal BCPN production and that other BBN derivatives also contributed to intestinal BCPN production. Indeed, untargeted metabolomics analysis detected glucuronidated BBN (gBBN) in the large intestine of ABX-treated animals, but not in animals without ABX treatment (Fig. 2d, Extended Data Fig. 2f and Supplementary Table 8). These data suggested deconjugation of gBBN in the caecum by β-glucuronidases that are prevalent in gut bacteria, followed by oxidation of liberated BBN to BCPN[16,17]. To test this hypothesis, we quantified caecal gBBN in both BBN-treated and ABX–BBN-treated mice following in vitro deglucuronidation of the samples as previously described[5,18]. Indeed, more BBN was liberated post-experimentally from large intestinal contents of mice co-treated with ABX and BBN compared to mice treated with BBN alone (Fig. 2e and Supplementary Table 8). Comparison of total BBN (free BBN and gBBN) and BCPN revealed that the amount of BCPN in BBN-treated animals was comparable to that of BBN in ABX–BBN-treated animals (Fig. 2f and Supplementary Tables 8 and 11), indicating that gut microbiota produces BCPN from both free BBN and gBBN following deglucuronidation. This hypothesis is further supported by the results of metabolic cage experiments demonstrating that more gBBN is eliminated in urine and faeces of ABX–BBN-treated compared to BBN-treated mice (Extended Data Fig. 2g,h and Supplementary Table 17). To directly test intestinal conversion of BBN to BCPN in the absence of deglucuronidation, we carried out an independent mouse experiment, in which we orally administered a single dose of BBN (0.05 mg per g (body weight)) to C57BL/6 mice either with or without 2 weeks of ABX pre-treatment. We found that despite comparable liver biotransformation of BBN to BCPN (Extended Data Fig. 2i,j and Supplementary Table 18), caecal BCPN levels were already elevated in the group without ABX pre-treatment within 1 h of BBN administration (*P* value = 0.085; Extended Data Fig. 2k and Supplementary Table 18), before biliary excreted gBBN reached the lower intestine[5] (no caecal gBBN was detected in either of the mouse groups 1 h after oral administration; Extended Data Fig. 2l and Supplementary Table 18). These data suggested that BCPN can be directly produced through the oxidation of BBN in the intestine. Another potential substrate for generating BCPN in the intestine is gBCPN through bacterial deglucuronidation. We indirectly quantified gBCPN by calculating the additional BCPN released after in vitro deglucuronidation (Fig. 2e). However, no significant differences in BCPN levels before and after deglucuronidation could be detected in either of the two groups (Extended Data Fig. 2m and Supplementary Table 10), suggesting that gBCPN is a neglectable metabolite in the gut, whose deglucuronidation cannot account

for the difference in overall BCPN levels between the groups. Overall, these observations corroborate our findings that intestinal BCPN is produced in a microbiota-dependent way through the combination of deconjugation of gBBN and oxidation of BBN, which leads to altered toxicokinetics with increased urinary elimination and urinary tissue accumulation of the carcinogenic BCPN.

## Gut bacteria oxidize BBN to BCPN

Having found that the observed microbiome-dependent differences in BCPN are caused by intestinal deconjugation of gBBN and subsequent metabolism of BBN, we sought to test whether gut microorganisms could directly oxidize BBN to BCPN. To this end, we cultured mouse caecal, colon and rectal samples under anaerobic, microaerobic (10% $O_2$) and aerobic (21% $O_2$) conditions, and incubated these ex vivo microbiota cultures with BBN. Quantification of BCPN after 4, 8, 12 and 24 h of BBN exposure demonstrated the capacity of these microbial communities to oxidize BBN to BCPN solely under aerobic and microaerobic atmospheres despite comparable final bacterial densities between all conditions (Fig. 3a, Extended Data Fig. 3a and Supplementary Table 19). Although oxygen dependency of BBN conversion to BCPN is not unexpected given the oxidative nature of the reaction, the fact that we also observed this reaction in the large intestine of mice, which is thought to be mostly anaerobic, is notable. Altogether, these results demonstrate that gut microbial communities have the metabolic capacity to generate BCPN from BBN.

To identify gut bacterial species that can convert BBN to BCPN, we assembled a collection of 564 axenic bacterial cultures isolated from the mouse intestine under aerobic and microaerobic conditions and tested them for BBN oxidation under accordant growth conditions. Using DNA-sequencing analysis of the full-length 16S rRNA gene of active isolates, we identified 12 species belonging to 8 different bacterial genera (that is, *Escherichia, Pseudolabrys, Lactobacillus, Staphylococcus, Corynebacterium, Kocuria, Arthrobacer* and *Micrococcus*) and three distinct phyla (Firmicutes, Proteobacteria and Actinobacteria; Supplementary Table 21). To filter for species that are present in the mouse microbiome at detectable levels and to account for potential isolation biases due to (micro)aerobic enrichment culturing, we carried out 16S rRNA sequencing of caecal, colon and rectal samples. We then mapped the reference sequence of the amplicon sequence variants (ASVs) identified in the samples to full-length 16S rRNA sequences of the BBN-converting bacterial isolates. This identified eight ASVs belonging to four different genera (*Escherichia, Lactobacillus, Corynebacterium* and *Staphylococcus*; Fig. 3b, Extended Data Fig. 3b,c and

Supplementary Table 22). This approach further allowed us to estimate the collective abundance of these putative BBN-metabolizing bacteria in the gut, which we found to be low (0.48% in caecum, 1.66% in colon and 0.50% in rectal contents; Fig. 3c and Supplementary Table 22). These observations suggest that BBN oxidation is performed by a distinct subpopulation of the gut microbiota, as expected given the oxygen dependence of the reaction in a mostly anaerobic environment.

## Human gut microbiota convert BBN to BCPN

Next we investigated whether microbial communities of the human gut could also oxidize BBN to BCPN, despite generally large compositional differences compared to the mouse gut microbiome[19,20]. To this end, we collected 11 culturable faecal samples from human volunteers and carried out 16S rRNA sequencing analysis. As expected, these gut microbial communities collectively shared only 10.56% of the exact taxonomy and 3.36% of exact ASVs with the mouse communities, with only one putatively active bacterial genus (*Escherichia*) overlapping (Supplementary Table 25). To directly assess the BBN-converting capacity of human gut microbiota, we cultured these faecal communities ex vivo under anaerobic, microaerobic and aerobic conditions, incubated them with BBN and quantified BCPN production after 24 h as described above. Despite comparable culture density between the communities, which we assessed by colony-forming unit counting (Extended Data Fig. 4a and Supplementary Table 24), only six of ten communities (one community was excluded owing to technical reasons) produced BCPN after 24-h BBN incubation with the strongest activity under microaerobic conditions (Fig. 4a and Supplementary Table 23). Additionally, these six active microbial communities showed strong differences in BBN-oxidation activity, suggesting pronounced person-to-person differences in microbiota BBN metabolism. To attempt to explain these interindividual differences, we explored the overall difference in microbiome composition between individuals, but did not find any grouping based on the BBN conversion capacity of the faecal communities (permutational multivariate analysis of variance test statistic 0.680505, *P* value 0.613, 999 permutations; Extended Data Fig. 4b). Overall, these results demonstrate that, similar to the mouse gut microbiota and despite distinct compositional differences, certain microbial gut communities from humans are also capable of oxidizing BBN to BCPN.

To further confirm the capacity of human microbiota to convert BBN to BCPN in vivo, we colonized germ-free mice with the stool microbiota of participant G11 before carrying out the same BBN-exposure experiment as described above for conventional animals with these 'humanized' mice (Fig. 1a). Intestinal quantification of BBN and BCPN reproduced the metabolic phenotype observed in conventional mice and also demonstrated intestinal BCPN production by human gut microorganisms (Fig. 4b, Extended Data Fig. 4c–f and Supplementary Tables 26–31).

To identify human gut bacteria converting BBN to BCPN, we first tested 16 different gut bacterial type strains, covering the four most abundant phyla in the gut, for their capacity to convert BBN to BCPN in vitro under various growth conditions at 0%, 10% and 21% $O_2$. Of these tested strains, only *Escherichia coli* ED1a could produce BCPN (Fig. 4c and Supplementary Table 32), corroborating our finding of *Escherichia* being the only BBN-converting genus of the mouse microbiome also found in the human microbiome. To identify additional BBN-metabolizing human gut bacteria not included in the panel of type strains, we assembled personalized culture collections of a total of 430 axenic bacterial cultures isolated from the different faecal communities under microaerobic and aerobic conditions. We then tested each of the isolates for its ability to oxidize BBN, resulting in the identification of up to 15 BBN-converting species from each faecal community (Supplementary Table 33). In total, we isolated 25 unique BBN-converting species and 18 different bacterial genera from 2 distinct

phyla (Firmicutes and Proteobacteria; Extended Data Fig. 4g). Notably, several BBN-converting species could be isolated from faecal communities that were not active BBN converters as a whole in our community BBN conversion assay. This may be explained by the generally low abundance of these BBN-converting bacterial strains in the communities, limited oxygen availability in the dense community context due to competition between (micro)aerobes, and different bacterial physiology (that is, gene expression) between axenic and community cultures. To identify bacterial isolates that are at detectable levels in the original human faecal samples and to account for potential cultivation biases due to (micro)aerobic enrichment culturing, we mapped the full-length 16S rRNA sequences of the isolates to the 16S rRNA microbiome data, as described above. We were able to map a total of seven ASVs of the faecal microbiomes, belonging to three different genera (*Escherichia*, *Enterococcus* and *Hemophilius*; Extended Data Fig. 4h and Supplementary Table 33). Similar to that of the mouse microbiota, relative abundance of these ASVs (putative BBN metabolizers) ranged between 0% and 10.52% with an average of 3.94% (Extended Data Fig. 4i) in the different communities. However, despite the large differences in the relative abundance of putative BBN-converting ASVs, we did not find any correlation between the total relative abundance of these ASVs and the BBN conversion capacity of faecal communities (Extended Data Fig. 4j). This may be due to the fact that the community context matters for BBN oxidation or due to the limited resolution of 16S rRNA amplicon sequencing, which could not fully reflect species- and strain-level BBN metabolism differences. Although the relative abundance of these bacteria could not predict a microbial community's metabolic potential to oxidize BBN, our data demonstrate that there are marked interpersonal differences in gut microbiota BBN oxidation capacity and that distinct (micro)aerobic human gut microorganisms with the ability to metabolize BBN can be isolated.

The fact that we could isolate BBN-converting bacterial species unique for a person's microbiome or shared between multiple individuals and the absence of a clear correlation between bacterial taxa and community BBN metabolism led us to investigate whether BBN metabolism differs at the bacterial strain level. For example, we could isolate *Escherichia* isolates from different individuals (and from mice; Figs. 3b and 4d), suggesting that the ability to convert BBN to BCPN is conserved across different strains of this bacterial genus. To test this hypothesis, we sequenced genomes of seven *Escherichia* isolates from different people, demonstrating that they are indeed different strains (Extended Data Fig. 4k and Supplementary Tables 34 and 35). Additionally, we tested 95 diverse *E. coli* strains previously isolated[21], for their capacity to produce BCPN from BBN (Fig. 4e and Supplementary Table 36). We found that all of the tested strains were capable of metabolizing BBN, suggesting that the capacity to oxidize BBN is shared among *E. coli* strains and common to different *Escherichia* species. To test BBN metabolism of a representative human *Escherichia* isolate in vivo, we monocolonized germ-free animals with one of our isolates (*Escherichia flexneri*) to compare intestinal BBN metabolism between these monocolonized and germ-free animals following 10 days of BBN administration in drinking water. Measurements of both gBBN and BCPN recapitulated our finding that gut bacteria metabolize BBN in the intestine of conventional and 'humanized' mice, leading to BCPN accumulation and absorption in the lower intestine (Fig. 4f,g and Supplementary Tables 37 and 38). Furthermore, to better recapitulate the interspecies interactions in the gut, we also inoculated germ-free mice with a two-member community consisting of the bacterial type strains *Coprococcus comes* and *Flavonifractor plautii* that we found to not metabolize BBN (Fig. 4c) and a three-member community with the isolated BBN-metabolizing *E. flexneri* strain added. Without differing in host metabolism (Fig. 4h), the three-member bacterial community led to BCPN accumulation in the lower intestine, in contrast to the two-member community that lacked BBN-metabolizing bacteria (Fig. 4i,j and Supplementary

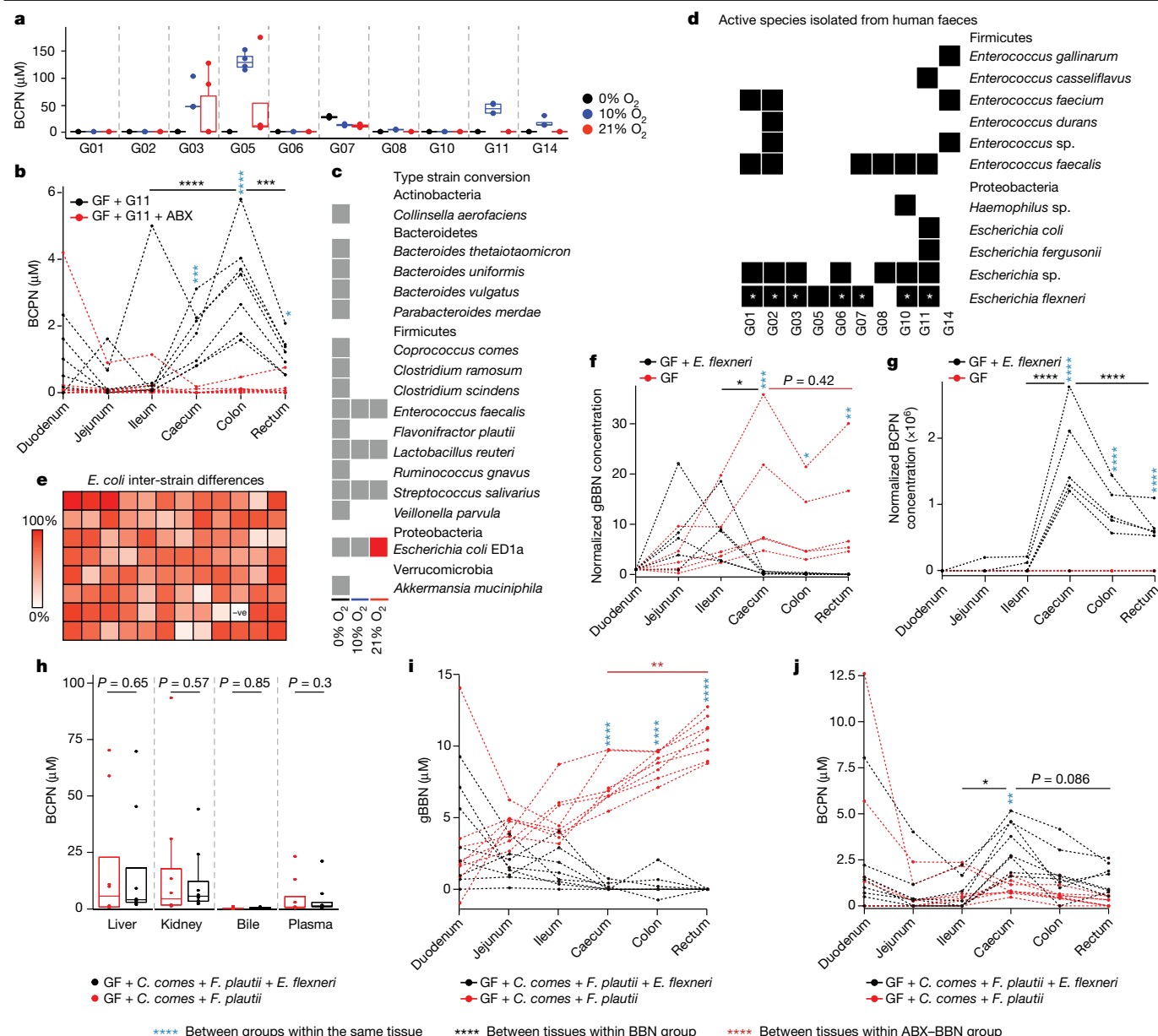

**Fig. 4 | Interpersonal differences in BBN-to-BCPN conversion and metabolic activities of single isolates in vitro and in gnotobiotic mouse models.**
**a**, BCPN production of ex vivo faecal cultures from humans after 24 h of BBN incubation under anaerobic (black), microaerobic (blue) and aerobic (red) conditions (*n* = 4 technical replicates; Supplementary Table 23). **b**, Intestinal BCPN of germ-free (GF) mice colonized with faecal microbiota from participant G11 following oral BBN or ABX–BBN administration for 10 days (*n* = 8 and 7; Supplementary Table 26). Comparisons between groups were carried out with EMMs after fitting the data with a linear mixed model, and *P* values were calculated using a two-sided *t*-test and adjusted for multiple testing with FDR (Supplementary Table 29). **c**, BBN-to-BCPN conversion of human gut bacteria after 24-h BBN exposure. Grey squares indicate assays carried out under given oxygen conditions; coloured squares indicate BCPN production (Supplementary Table 32). **d**, Bacterial isolates from communities from **a** capable of BBN-to-BCPN conversion. Asterisks indicate isolates subjected to whole-genome sequencing and sequence comparison (Extended Data Fig. 4k and Supplementary Table 33). **e**, BBN-to-BCPN conversion by 95 *E. coli* strains[21]

(Supplementary Table 36). Colour intensity indicates relative production of BCPN after 24 h of BBN incubation. −ve, no-bacteria control. **f**,**g**, Intestinal concentrations of gBBN (**f**) and BCPN (**g**) in *E. flexneri* monocolonized mice after 10 days of BBN administration, normalized by respective concentrations in duodenum (Supplementary Table 37). Pairwise comparisons between the same tissue of different groups and between tissues in the same group were carried out with EMMs after fitting the data with a linear mixed model, and *P* values were calculated using a two-sided *t*-test and adjusted for multiple testing using FDR (Supplementary Table 38). **h**–**j**, BCPN (**h**,**j**) and gBBN (**i**) concentrations in liver, kidney, bile and plasma (**h**) and intestine (**i**,**j**) of germ-free mice colonized with two-member or three-member bacterial communities (*n* = 8 and 8, two-member and three-member group, two-sided Wilcoxon test; Supplementary Tables 39 and 40). *\*P* ≤ 0.05, *\*\*P* ≤ 0.01, *\*\*\*P* ≤ 0.001, *\*\*\*\*P* ≤ 0.0001. For all box plots, the lower and upper hinges depict the 25th and 75th percentiles. The upper and lower whiskers extend from the hinge to the largest and smallest values at most 1.5 × IQR from the hinge. The centre line in the box shows the median.

Tables 39 and 40). Overall, these results demonstrate that specific human gut bacteria are capable of metabolizing BBN in the intestine, which leads to altered toxicokinetics of the nitrosamine carcinogen.

## Microbiota affects other nitrosamines

Having established that the toxicokinetics of BBN is altered by gut microbial metabolism, we questioned whether related nitrosamine

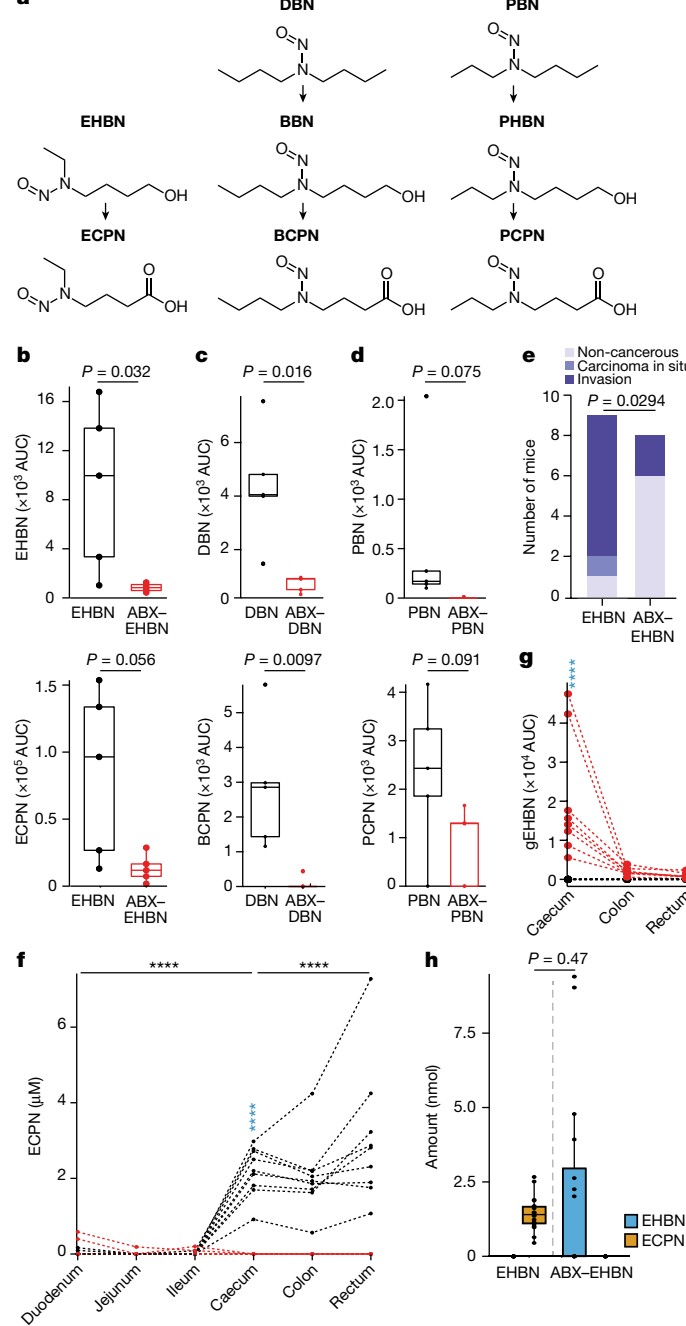

**Fig. 5 | Microbiota-dependent toxicokinetics of BBN-related nitrosamine carcinogens. a**, Chemical structures of EHBN, DBN and PBN and their oxidized metabolites, ECPN, propyl-(4-hydroxybutyl)-nitrosamine (PHBN) and *N*-propyl-*N*-(3-carboxypropyl)-nitrosamine (PCPN). **b**–**d**, Urinary excretion of the three nitrosamines and their metabolites over 24 h collected in metabolic cages. The *P* values between ABX-treated and ABX-untreated mice (*n* = 5 and 5) were calculated with a two-sided Wilcoxon test (Supplementary Table 41). AUC, area under the curve. **e**, Number of mice fed on EHBN alone or together with ABX at each bladder cancer development stage after 20 weeks of treatment (*n* = 9 and 8, two-sided $\chi^2$ test with Yates continuity correction; Supplementary Table 42). **f**,**g**, ECPN (**f**) and glucuronidated EHBN (gEHBN; **g**) levels in the gastrointestinal tract of the two groups of mice at 8 weeks of EHBN treatment (Supplementary Tables 43 and 45). Pairwise comparisons between groups for the same tissue and between tissues in the same group were carried out using EMMs after fitting the data with a linear mixed model, and *P* values were calculated using a two-sided *t*-test and corrected for multiple testing using FDR (Supplementary Tables 44 and 46). **h**, Mass balance between EHBN level in caecum of mice treated with EHBN alone and ECPN level in mice treated with both EHBN and ABX (*n* = 9 and 8; Supplementary Table 47). Pairwise comparisons between EHBN and ECPN in different groups were carried out with EMMs and *P* values were adjusted with the Tukey method (Supplementary Table 48). ****$P \leq 0.0001$; colours as in Figs. 2 and 4. For all box plots, the lower and upper hinges depict the 25th and 75th percentiles. The upper and lower whiskers extend from the hinge to the largest and smallest value at most $1.5 \times IQR$ from the hinge. The centre line in the box shows the median.

for 20 weeks. Consistent with our findings for BBN, our observations also revealed bladder carcinogenesis in EHBN-exposed mice with intact microbiota, whereas we observed reduced tumorigenesis in ABX–EHBN-treated animals ($\chi^2$, *P* value = 0.029; Fig. 5e, Supplementary Data 2 and Supplementary Table 42). Additionally, we observed that *N*-ethyl-*N*-(3-carboxypropyl)-nitrosamine (ECPN), the oxidized metabolite of EHBN, accumulated in the large intestine only in non-ABX-treated mice, similar to BCPN accumulation following BBN exposure (Fig. 5f, Extended Data Fig. 5a and Supplementary Tables 43 and 44). Finally, measurements of glucuronidated EHBN and mass balancing total EHBN in ABX–EHBN-treated mice with ECPN in EHBN-treated mice demonstrated that the observed alteration in toxicokinetics of EHBN is probably the combination of gut bacterial deconjugation and oxidation reactions (Fig. 5g,h and Supplementary Tables 45–48). Altogether, these results indicate that gut microbial metabolism can alter the toxicokinetics of nitrosamine carcinogens with effects on chemically induced tumorigenesis.

## Discussion

In this study, we demonstrated that microbial biotransformation of environmental carcinogens is a contributing factor for cancer development. We mechanistically demonstrate how the gut microbiota promotes chemically induced tumour development and accelerates cancer progression in a distal organ through microbial metabolism of environmental carcinogens. By inoculating active human faecal communities as well as single-strain isolates and synthetic bacterial communities with or without an active BBN converter strain into germ-free mice, we could recapitulate the toxicokinetics of microorganisms in converting BBN and its derivatives into BCPN as in conventional mice. We demonstrated the role of bacterial metabolism in not only regenerating the parent carcinogen through deglucuronidation, but also actively producing the carcinogenic metabolite in a two-step mechanism. We found pronounced differences in the capacity of individuals' microbiota to metabolize the carcinogen, suggesting that there exist interpersonal differences in microbiota contributions to chemical-induced tumour development. However, the abundance of BBN-metabolizing species in the microbiome could not explain the

carcinogens would be similarly affected by the gut microbiota. To this end, we carried out 24-h metabolic cage experiments with three additional nitrosamine carcinogens (*N*-ethyl-*N*-(4-hydroxybutyl)-nitrosamine (EHBN), *N*-nitrosodibutylamine (DBN) and *N*-propyl-*N*-butyl-nitrosamine (PBN)), with conventional and ABX-treated animals (Fig. 5a). As expected from our findings with BBN and BCPN, we found that the microbiota also altered the toxicokinetics of these additionally tested nitrosamine carcinogens, resulting in reduced urinary secretion of the nitrosamine and/or their oxidized metabolites following reduction of the gut bacterial load through ABX treatment (Fig. 5b–d and Supplementary Table 41).

As EHBN is commonly used to study chemical-induced bladder tumorigenesis in mouse models[22], we investigated whether the observed impact of the microbiota on toxicokinetics would also functionally translate into altered bladder tumorigenesis. To this end, we carried out the long-term exposure experiment with EHBN

carcinogen-metabolizing activity of the respective microbial community. This may be due to the fact that community context could determine carcinogen-metabolizing activities of individual bacterial strains and suggests further studies to better understand microbiota metabolism of environmental pollutants.

Together, these results complement previous studies implicating the gut microbiome in cancer development and provide a methodological and conceptual framework to study microbiome carcinogen metabolism and its consequences on toxicokinetics and tumour development. Future investigations will build on these findings aiming at identifying microbiome-encoded predisposition risk factors for chemical-induced cancer development and to enable rational strategies aimed at manipulating individuals' microbiota for cancer prevention.

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

## Methods

### Animal experiments

Two-month-old C57BL/6J mice (Jackson Laboratory and Charles River) housed in the animal facility of the University of Split School of Medicine were used for all conventional mouse experiments. Two-month-old germ-free C57BL/6J mice housed in the animal facility of the European Molecular Biology Laboratory (EMBL) were used for all gnotobiotic mouse experiments. Animals were kept in individually ventilated cages with autoclaved bedding (Mucedola), with a 12-h light–dark cycle, controlled temperature (21–24 °C) and humidity (30%–70%), and ad libitum access to food (γ-irradiated, Ssniff-Spezialdiäten) and autoclaved water. Male mice were used in all experiments unless specified otherwise. All mouse experiments were approved by the local institutional animal care and use committee (The University of Split, School of Medicine, Animal Welfare Committee and EMBL Institutional Animal Care and Use Committee) and the national regulating authorities (Republic of Croatia Ministry of Agriculture, Veterinary and Food Safety Directorate; permit numbers 525-10/0255-14-4, 525-10/0255-15-5, 525-10/0543-21-8, 525-09/566-22-2 and 21-002_HD_MZ). All mice were randomly allocated into experimental groups; blinding was not carried out, except for the pathohistological analysis. No sample size calculation was carried out. The general status of the animals including behaviour, appearance and body weight was monitored. If signs of pain, suffering or weight loss of more than 20% were observed, humane end points were applied. Those limits were not exceeded in any of the experiments. For ex vivo BBN conversion experiments and 16S rRNA sequencing using conventional mice, contents from the three parts of the small intestine, caecum, large intestine and rectum were collected from five mice of each sex. One half of each sample was frozen at −80 °C in 20% glycerol, and the other half was frozen at −80 °C and used for 16S rRNA amplicon sequencing. For liquid chromatography-coupled mass spectrometry (LC–MS) quantification, tissues collected were directly frozen at −80 °C until further processing.

### Chemicals

Chemicals used in this study are listed in Supplementary Table 49.

### Long- and short-term nitrosamine treatments with conventional or germ-free mice

Male conventional mice were randomly assigned to the ABX–BBN or BBN group. The ABX–BBN group received 100 μg ml⁻¹ neomycin, 100 μg ml⁻¹ metronidazole, 50 μg ml⁻¹ streptomycin, 67.7 μg ml⁻¹ penicillin (all from Sigma-Aldrich) and 50 μg ml⁻¹ vancomycin (Pharma Swiss) in drinking water that was supplied fresh weekly. Following 2 months of treatment, the ABX mixture was changed to 1,000 μg ml⁻¹ streptomycin, 170 μg ml⁻¹ gentamicin (Krka), 125 μg ml⁻¹ ciprofloxacin (MCE) and 1,000 μg ml⁻¹ bacitracin (Sigma-Aldrich), to prevent development and overgrowth of resistant bacterial strains[23]. The ABX–BBN group received ABX 2 weeks before starting the BBN treatment, and were kept on ABX for the remainder of the experiment.

For gnotobiotic mouse experiments, male germ-free mice were inoculated by oral gavage with 100–250 μl microbial community culture or single bacterial isolate culture for 4 days. The ABX–BBN group would receive the first ABX mixture for 2 weeks before starting BBN treatment for 10 days. During BBN treatment, the second ABX mixture was used for the ABX–BBN group.

Monitoring of the impact of ABX treatment on the gut microbiota was carried out using a combination of colony-forming unit (CFU) counting and 16S rRNA amplicon sequencing (microbiome composition). For CFU counting, faecal pellets from both groups were diluted (1 g in 10 ml) in PBS and homogenized by vortexing. Decimal dilutions (10⁻¹ to 10⁻⁹) were prepared and grown on blood agar (TSA, 10% blood) at 37 °C under aerobic and anaerobic conditions prepared with gas packs (AnaeroGen; Thermo Fisher). CFUs were calculated using two successive dilutions. For 16S rRNA sequencing, faecal pellets were collected from mice and DNA was extracted using the PowerSoil Kit (QIAGEN). Sequencing was carried out on an Illumina platform generating 250-base-pair (bp) paired-end sequences of the V4 region.

After checking the effects of ABX, we gave both groups of mice the nitrosamines (0.05% BBN and 2.87 mM EHBN, DBN and PBN) in drinking water. BBN was administered for 12 weeks, after which the chemical was removed from the drinking water for 8 weeks. PBN was administered for 17 weeks, whereas EHBN and DBN were administered for 20 weeks[6,24].

For tissue collection, mice treated with BBN and EHBN were euthanized by inhalation of $CO_2$, and intestinal contents, liver, kidney, urinary bladder and plasma were collected and frozen at −80 °C until further analyses. Bladder tissues were cut in half longitudinally, half of which was frozen for metabolomics quantification and the other half was fixed for histopathology analyses.

For monitoring urinary and faecal excretion, 24-h urine and faecal samples from mice treated with all chemicals were collected using metabolic cages (Techniplast). Mice were placed inside cages supplied with food and water (supplemented with nitrosamines with or without ABX for respective groups) for 24 h and faecal material and urine was collected from the collection tubes as well as from the surfaces of the cages. Samples were weighed and stored at −80 °C until further analyses.

### Pharmacokinetics experiment

Fifty conventional mice of both sexes were randomly assigned to the ABX–BBN or BBN group. The ABX–BBN group was given a mix of ABX (neomycin, metronidazole, streptomycin, penicillin and vancomycin at the same concentrations as described above) for 2 weeks before receiving BBN. Both groups of mice were given a single dose of BBN (50 mg per kg body weight) by oral gavage and euthanized after 1, 3, 6 and 9 h (five mice per time point, per group) and intestinal contents, liver, gallbladder, kidneys and plasma were collected, weighed, snap frozen in liquid nitrogen and stored at −80 °C until further analysis.

### Collection and preparation of human microbiome samples

Patients were recruited from the Gastroenterology Department at University Hospital of Split between March and August 2020. The study was approved by the ethics committee of the University Hospital of Split (permit number 2181-147-01/06/M.S.-20-4) and the University of Split School of Medicine (permit number: Ur. br. 2181-198-03-04-20-00400) and all participants gave informed consent before participation in the study.

Faecal samples were collected from 12 patients undergoing gastroenterological analysis for different medical conditions. Exclusion criteria for all patients included in the study were: usage of ABX one month before sample collection and presence of malignant pathology of the analysed tissues.

Half of the sample volume was mixed with glycerol (20% final concentration) and stored at −80 °C to conserve viable microorganisms for subsequent in vitro BBN conversion assays. The remaining sample volume was directly stored at −80 °C for DNA extraction (PowerSoil Kit, QIAGEN) and 16S rRNA sequencing.

### Histological analysis

Following euthanization of mice, the bladders were collected and medially cut into equal halves using a scalpel. One half was immediately frozen in liquid nitrogen and kept for subsequent analysis. The other half was immersed in neutral-buffered formalin (10%) for 24 h. Following fixation, tissues were dehydrated using a series of ethanol dilutions (75%, 90%, 95% and three times 100%, for 1 h each), cleared with xylene (three series for 30 min each) and finally embedded in

paraffin (immersed in two series of paraffin for 1 h each and embedded in a third series of paraffin). Embedded tissues were then cut into 5-µm sections with a microtome (RM2125 RTS, Leica) and stained with haematoxylin (Sigma) and eosin (Merck) for microscopical examination. Blinded histological assessment was carried out by researchers and by a trained pathologist.

## HEP-G2 cell line experiment
HEP-G2 cells were treated at 70% confluence with either BBN (0.05%) or BBN (0.05%) plus ABX mixture (containing vancomycin, streptomycin, metronidazole, neomycin and penicillin (in the same concentrations as in the mouse experiment)) in DMEM medium (Sigma-Aldrich) with 10% FBS (PAA Laboratories) for 24 h. Cells were pelleted (300 rcf, 3 min) and supernatant was used for BBN and BCPN measurements by high-performance liquid chromatography (PerkinElmer series 200) as described previously[25]. HEP-G2 (ACC-180) cells were obtained from the DSMZ (German Collection of Microorganisms and Cell Cultures).

## Ex vivo and in vitro BBN conversion assays
CFU of ex vivo microbial communities were determined as described above. To isolate single ex vivo communities, bacterial cultures at different dilutions were streaked onto brain heart infusion (BHI) blood plates, and incubated under anaerobic, microaerobic and aerobic oxygen conditions and incubated at 37 °C for 24–48 h. Between 24 and 48 single colonies were then randomly picked from dilution plates when possible. The full-length DNA of the 16S rRNA gene of each isolate was PCR-amplified using the 27f (5′-AGAGTTTGATCATGGCTCA-3′) and 1492r (5′-TACGGTTACCTTGTTACGACTT-3′) primer pair and sent for Sanger sequencing (Eurofins; Supplementary Tables 50 and 51). Type strains from the German Culture Collection (DKFZ) or the American Type Culture Collection (ATCC) were streaked onto BHI blood plates and allowed to grow for 1–2 days before use.

Intestinal and faecal samples obtained from mice and humans were grown as ex vivo microbial communities and single bacterial isolates and strains were grown as axenic cultures at 37 °C under 0% (anaerobic, Coy chamber), 10% (microaerobic, Coy chamber) and 21% (aerobic, incubator) oxygen conditions in BHI-S medium (37.0 g BHI broth, 5 g yeast extract supplemented with L-cysteine HCl, haemin and vitamin K1 in 1 l distilled water) or modified GAM (MGAM) broth (HyServe, catalogue number 5433) for 2 h for communities to prevent overgrowth of strains in microbial communities or overnight for axenic cultures. Cultures were then treated with 10 µM BBN after mixing with an equal volume of twofold-diluted BHI-S medium to allow further exponential growth. A 20 µl volume of the treated culture was collected after 0, 4, 8, 12 and 24 h of BBN incubation and immediately frozen in a 96-well V-bottom storage plate (Fisher Scientific, catalogue number 10304513), and sealed with aluminium foil at −80 °C for storage until further processing. To ensure even oxygen distribution in each well under microaerobic and aerobic conditions, the assays were carried out on an orbital plate shaker (Thermo Shaker PHMP) at 600 r.p.m.

## 16S rRNA sequencing
DNA was isolated from mouse and human samples using the QIAGEN PowerSoil kit and quantified on Qubit (Thermo Fisher Scientific). DNA library preparation and sequencing were carried out as described before[26]. In brief, for the first PCR reaction, 5–20 ng of DNA was amplified for 25 to 30 cycles with KAPA or Q5 (Bio-Labs) master mix with primers for the variable region V4 of the 16S rRNA gene (forward: 5′-GTGCCAGCMGCCGCGGTAA-3′; reverse: 5′-GGACTACHVGGGTWTCTAAT-3′)[27]. For the second PCR reaction, 1 µl or 2 µl, depending on the PCR product band width on the gel, of the first PCR product was mixed with primers (NEXTFLEX 16S V1-V3 Amplicon-Seq Kit, PerkinElmer), and KAPA or Q5 (Biolabs) master mix

and amplified for 25 cycles. After pooling, the library was cleaned with the AMPure XS kit (Beckman) with 0.75× ratio according to the manufacturer's protocol. All samples were paired-end sequenced with 250-bp read lengths on the Illumina MiSeq platform at the Genomics Core Facility of EMBL Heidelberg.

Raw data quality was assessed using FastQC v0.11.5 and raw data were imported in QIIME2 v2020.8 for downstream analysis[28]. Samples were checked for adapter contamination with cutadapt[29] and passed to dada2[30] for denoising, dereplication and chimera filtering; a feature table describing the distribution of reads in each sample among the identified ASVs was created, together with a representative sequence for each of the ASVs. The representative sequences identified for each ASV were used to create a fragment insertion tree using sepp[31]. The feature table and the built insertion tree were used to compute α- and β-diversity metrics, separately for mice and humans.

DNA sequences of the whole 16S rRNA gene (V1–V9) from isolated bacteria obtained by Sanger sequencing were also imported into QIIME v2020.8, together with their corresponding taxonomy, as derived from the ACT service[32]. When possible, ACT taxonomy was further detailed at species level (if not already present) by directly aligning complete sequences to the 16S–/ITS reference database using BLAST[33], only if a percentage identity of >99% could be detected over an alignment length of minimum 700 bp. ASV representative sequences were aligned to these full-length 16S rRNA sequences using vsearch[34] with the following parameters: --p-maxaccepts 'all', --p-perc-identity 0.98, --p-query-cov 0.98, --p-top-hits-only True. Representative sequences not aligned to any Sanger sequence were assigned a taxonomy with a pre-fitted sklearn-based classifier[35], trained over the Greengenes 13_8 99% operational taxonomic unit full-length sequences. The feature table and taxonomy thus obtained were exported to plain tsv files, to be imported into R 4.0.0 (https://www.R-project.org/).

All ASVs found in mouse and human samples were respectively aligned with mafft[36] and used to construct a phylogeny with fasttree2[37]. The phylogenetic tree was exported in Newick format and imported into iTOL[38] for visualization and tree annotation.

## Whole-genome sequencing
Bacterial isolates were grown in liquid culture overnight, and DNA was extracted with PowerSoil kit (QIAGEN). Libraries for whole-genome sequencing were prepared using the MagicPrep NGS system with 40 µl of 100 ng DNA. High-throughput sequencing was carried out with MiSeq Reagent Kit v2 Micro, resulting in 155-bp-long paired-ends reads. Sample quality was assessed using FastQC v0.11.9[39] and MultiQC 1.12[40]; sample reads were trimmed for both quality and length using Trimmomatic 0.39[41] with the following options: removal of TruSeq adapters sequences; sliding window trimming, clipping the read once the average quality within the window (4 bp) falls below 20; finally, drop the read if it is shorter than 38 bp (Supplementary Table 35). Subsequently, reads were de novo-assembled using Spades v3.15.3[42], and the quality of assembly was assessed using QUAST v5.0.2[43].

Three different approaches were used to assess species assignation: a k-mer-based approach, using kraken v2.0.7[44]; a marker gene approach, using gtdbtk v2.1.1[45]; a 16S-based approach, using mTAGs v1.0.4[46]. All three approaches agreed in classifying all isolates as *Escherichia* or *Escherichia–Shigella*.

All assemblies were therefore annotated using Prokka v1.13[47] with the genus *Escherichia* as Organism details. Annotation results were passed to Roary v3.7.0[48] for pangenomic analysis. The 7 isolates shared 3,277 core genes over a total of 7,606 genes (Supplementary Table 36). Pangenome results were visualized with Phandango v1.3.0[49], and the tree built from accessory genome presence absence was visualized using iTOL v6[50].

## Sample preparation for LC–MS measurements

Sample preparation for LC–MS measurements was carried out as previously described[5]. In brief, for liquid tissues (that is, urine and plasma), 20 μl of the samples was subjected to a freeze–thaw cycle before the addition of 100 μl of acetonitrile/methanol (1:1), and 5 μl of 8 μM of internal standard, warfarin, was added to each sample. Samples were incubated at −20 °C for 30 min before centrifugation at 4 °C at 4,500 r.p.m. for 15 min. A 10 μl volume of the supernatant was diluted with 10 μl of water before LC–MS measurements. For solid tissues (that is, intestine, kidney, liver and bladder), samples were homogenized with 3-mm tungsten carbide beads (QIAGEN, catalogue number 69989) in 300–400 μl of acetonitrile/methanol (1:1) containing 320 nM internal standard by 5 min bead-beating at 30 Hz on a QIAGEN TissueLyser II. After 30 min incubation at −20 °C and 15 min centrifugation at 4 °C and 10,000 rcf, 10 μl of the supernatant was diluted with water at 1:1 or more before LC–MS measurements.

## Quantification of BBN and BBN metabolites by LC–MS

Chromatographic separation was carried out on a 3.0 mm × 10 cm Poroshell120 HPHC18 column with 1.9 μM particle size (Agilent Technologies) mounted on an Agilent 1290 Infinity II LC system coupled to a 6550 iFunnel qToF mass spectrometer. The column temperature was maintained at 45 °C. The mobile phase was composed of A: water with 0.1% formic acid; and B: methanol with 0.1% formic acid. A 5 μl volume of each sample was injected at 0.6 ml min$^{-1}$ flow rate starting from 5% mobile phase B followed by a linear gradient to 95% over 5.5 min. The column was allowed to re-equilibrate to starting conditions for 1.1 min before each sample injection. The mass spectrometer was operated in both negative and positive scanning mode (50–1,700 $m/z$) with the following source parameters: VCap, 3,500 V; nozzle voltage, 2,000 V; gas temperature, 275 °C; drying gas, 13 l min$^{-1}$; nebulizer, 45 psi; sheath gas temperature, 275 °C; sheath gas flow, 12 l min$^{-1}$; fragmentor, 365 V. Online mass calibration was carried out using a second ionization source and a constant flow (2 ml min$^{-1}$) of reference mass solvent (purine $m/z$ = 121.0509 and hexakis $m/z$ = 922.0098 for positive mode, betaine $m/z$ = 112.9856 and hexakis $m/z$ = 1033.9881 for negative mode). Standard curves for BBN and BCPN were obtained by serial-diluting each compound in water at twofold from 10 μM to 4.9 nM.

The MassHunter Qualitative Analysis software (Agilent Technologies, v10.0) was used to determine the retention time for all compounds to enable targeted analysis and quantification. Peak integration was carried out in MassHunter Quantitative Analysis software (Agilent Technologies, v10.0) with the following settings: mass tolerance = 20 ppm; peak filter at signal-to-noise ratio = 2; and retention time tolerance of 0.5 min. All statistical analyses and plotting were carried out in RStudio (v4.2.0) after exporting the data files from MassHunter Quantitative Analysis software. BBN and BCPN concentrations were calculated using the recorded calibration curves and the signals of the internal standard. In brief, the area of the internal standard warfarin in each sample was divided by the median area of warfarin across all samples in the same tissue to obtain the correction factor for each sample, which was then used to normalize the areas for the targeted compounds in each sample. Total drug amounts in each intestinal compartment were calculated using the corresponding total sample weight.

## Reporting summary

Further information on research design is available in the Nature Portfolio Reporting Summary linked to this article.

## Data availability

Raw sequencing data have been deposited on the European Nucleotide Archive server, with the accession number PRJEB43561. The pre-fitted sklearn-based classifier, trained over the Greengenes 13_8 99% operational taxonomic unit full-length sequences, was obtained from https://data.qiime2.org/2022.11/common/gg-13-8-99-nb-classifier.qza (see https://resources.qiime2.org). Raw data for the LC–MS analysis have been deposited in the Metabolights public repository with the accession number MTBLS3581. All other data are available in the main text or the Supplementary Information.

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

**Acknowledgements** We thank the members of the laboratories of J.T. and M.Z. for helpful discussions, G. E. Maftei for technical support during gnotobiotic animal experiments carried out at the animal facility at EMBL, V. Benes and his team for technical assistance in genomic sequencing and A. Koumoutsi and A. Typas for sharing the E. coli natural strain collection. The laboratory of J.T. is supported by INGRA-DET and I. Đikić. This work was supported by Croatian Science Foundation grant number IP–2020-02-8921 to J.T. and the Daimler Benz Foundation, the Lung Cancer Research Foundation and the European Molecular Biology Laboratory (to M.Z.). M.Z. is an ERC investigator.

**Author contributions** J.T., M.Z., B.R. and B.Z. conceived the study. B.R., B.Z. and V.B.-P. conducted experiments with mouse models. B.R., K.V. and I.K. carried out histopathology experiments. B.Z. carried out LC–MS sample processing, measurements and data analyses. B.R., L.S., I.Lj. and E.Ć. carried out high-performance liquid chromatography measurements. B.R., B.Z. and E.M. carried out 16S rRNA amplicon analysis, B.R., B.Z. and A.K. carried out

bacterial community culturing and B.Z. carried out type strain culturing, under different oxygen levels. B.Z. and E.M. carried out bacterial strain isolation and identification. B.Z., B.R. and E.M. visualized the data. A.M., E.L.V. and Ž.P. recruited study participants. B.R., B.Z., E.M., J.T. and M.Z. wrote the manuscript. All authors approved the final draft of the manuscript.

**Funding** Open access funding provided by European Molecular Biology Laboratory (EMBL).

**Competing interests** The authors declare no competing interests.

**Additional information**
**Correspondence and requests for materials** should be addressed to Janoš Terzić or Michael Zimmermann.

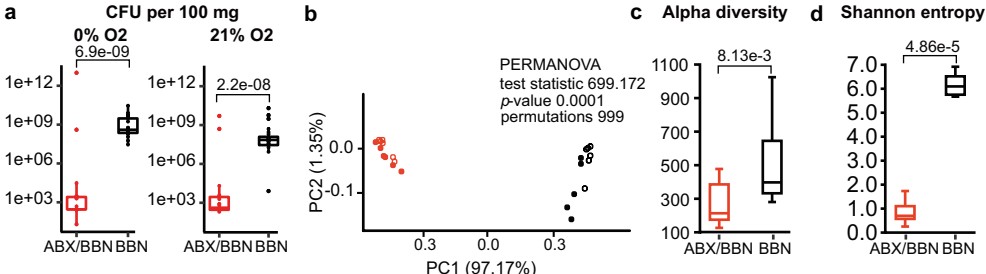

**Extended Data Fig. 1 | Fecal microbiota characterization after antibiotic treatment. a**. Colony forming unit (CFU) per 100 mg mouse feces with (red) and without (black) antibiotics treatment after 0, 2, 3, 8 and 12 weeks combined (n = 30/25, ABX-treated/untreated mice, two-sided Wilcoxon test, Table S1). **b**. Principal coordinates analysis of gut microbiome of ABX/BBN (red, n = 12) and BBN groups (black, n = 11) of mice (Table S2). Two-sided permanova test was used for partitioning distance matrices among sources of variation using permutations to obtain the *p*-values. White-filling: mice before BBN administration, solid filling: mice after 2 weeks of BBN administration. **c-d**. Alpha diversity (c) and Shannon entropy (d) of observed features in feces of ABX/BBN group (red) and BBN group (black). (n = 12/11) Kruskal-Wallis test was used to obtain the *p*-values. For each box plot, the center line represents the median, the box limits represent the 25th and 75th percentile, and the whiskers show 1.5× the interquartile range.

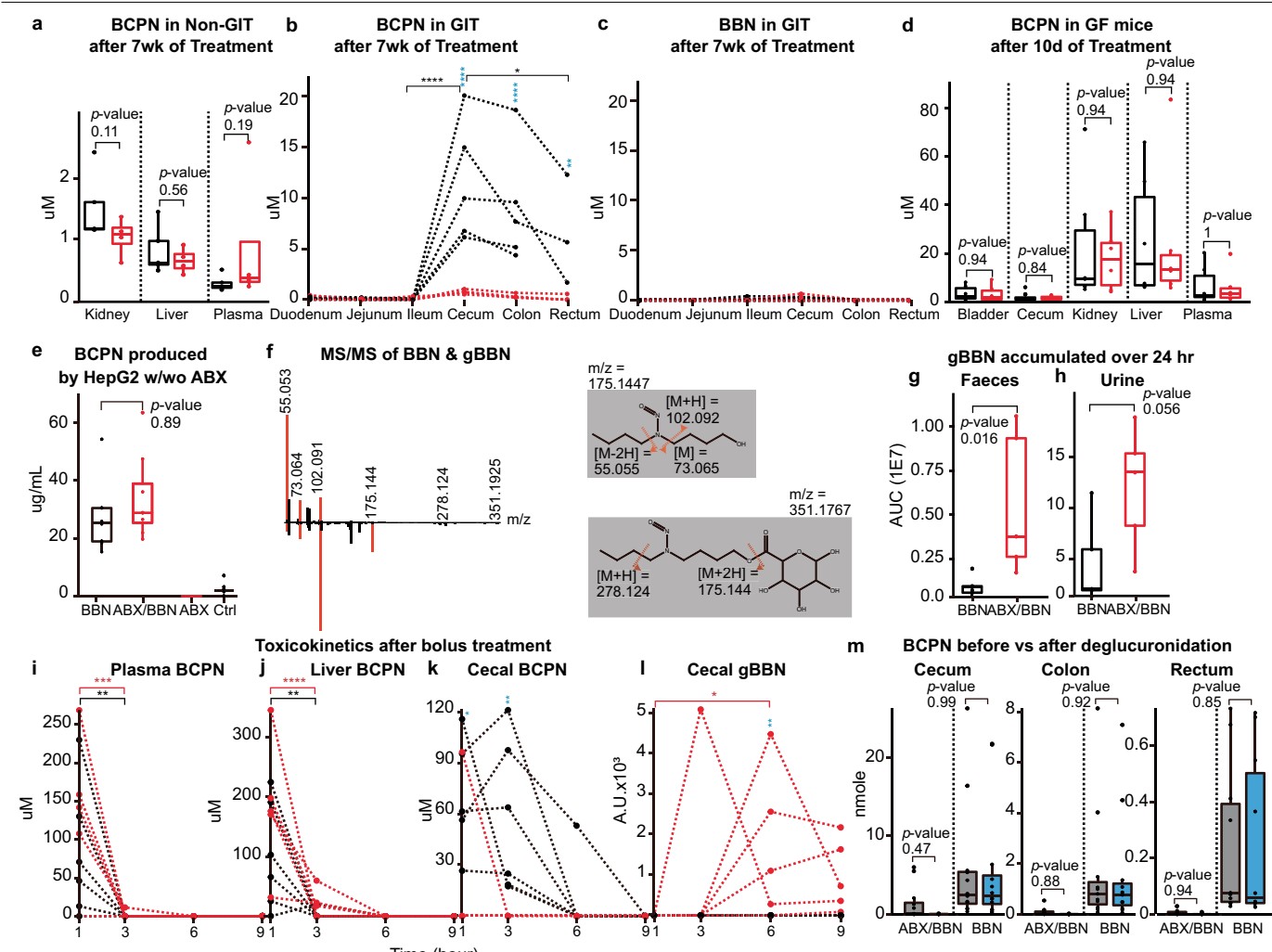

**Extended Data Fig. 2 | Additional long-term and one-bolus toxicokinetic measurements of BBN and its metabolites. a.** BCPN concentration after 7 weeks of BBN administration via drinking water in kidney, liver, and plasma in ABX/BBN (red) and BBN (black) mice (n = 4/5, two-sided Wilcoxon test, Table S12). **b-c.** BCPN (b) and BBN (c) concentrations along the gastrointestinal tract of ABX/BBN (red) and BBN (black) mice after 7 weeks of BBN administration via drinking water (Table S13). **d.** BCPN concentrations in bladder, cecum, kidney, liver, plasma in germ-free mice with (red) or without (black) antibiotic co-treatment followed by 10 days of BBN administration (n = 6/6, two-sided Wilcoxon test, Table S15). **e.** BCPN concentrations in HEP-G2 culture medium after exposure to 0.05% BBN at 70% confluency with ('ABX/BBN') or without ('BBN') antibiotic co-exposure (n = 8/8 technical replicates, two-sided Wilcoxon test). Antibiotics alone and cell culture alone (n = 7) were used as negative

controls (Table S16). **f.** Tandem MS/MS confirmation of BBN (above axis) and gBBN's (below axis) structures. **g-h.** gBBN amounts in excreted feces (g) and urine (h) over 24 h after 3 weeks of BBN exposure (n = 5/5, two-sided Wilcoxon test, Table S17). **i-l.** BCPN concentration in liver, plasma and cecum and gBBN concentration in cecum over 9 h after one-bolus oral gavage of 0.05 mg/g BBN. **m.** BCPN levels in cecum, colon and rectum before and after in vitro β-glucuronidase treatment after 3 weeks of BBN treatment (Table S8). Pairwise comparisons were performed with estimated marginal means (EMMs) after fitting the data with a linear mixed model, and p-values were calculated using a two-sided t-test and adjusted for multiple testing using FDR, **b-c** (Table S14), **i-l** (Table S18) and **m** (Table S10). Box plots: center line represents the median, box limits represent the 25th and 75th percentile, and whiskers show 1.5 × the interquartile range. p-values: 0.0001 ≤ ****, 0.001 ≤ ***, 0.01 ≤ **, 0.05 ≤ *.

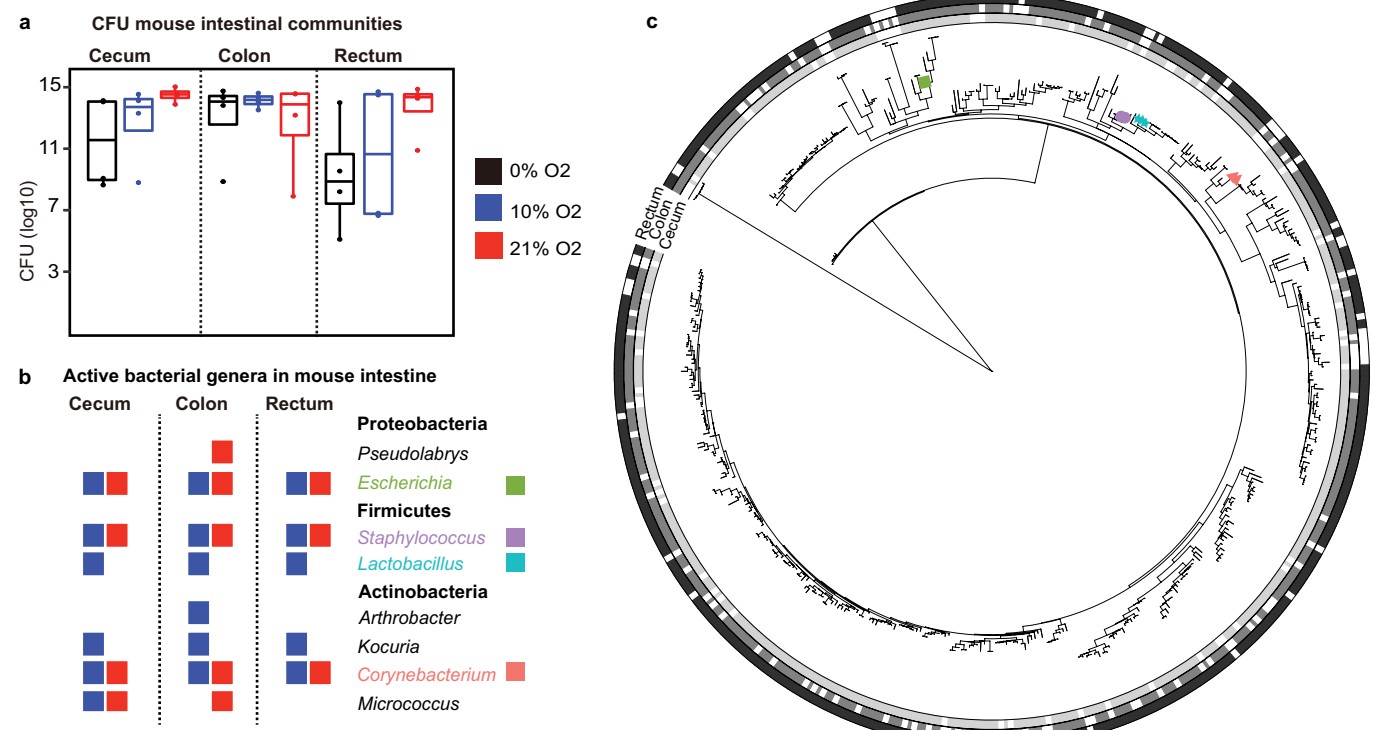

**Extended Data Fig. 3 | BBN-metabolizing bacterial species in the murine gut microbiome. a**. Colony forming units (CFU) of ex vivo cultures of samples collected from cecum, colon and rectum of mice after 24 h of incubation under anaerobic (black), microaerobic (blue) and aerobic (red) conditions (n = 8, 4 biological replicates with 2 technical replicates each, Table S20). Box plots: the center line represents the median, the box limits represent the 25th and 75th percentile, and the whiskers show 1.5× the interquartile range. **b**. Bacterial genera isolated from intestinal contents in **a** that converted BBN to BCPN under microaerobic (blue) and/or aerobic (red) conditions (Table S21). **c**. Mapping of the active genera identified in **b** in the same microbiomes characterized by 16S rRNA sequencing.

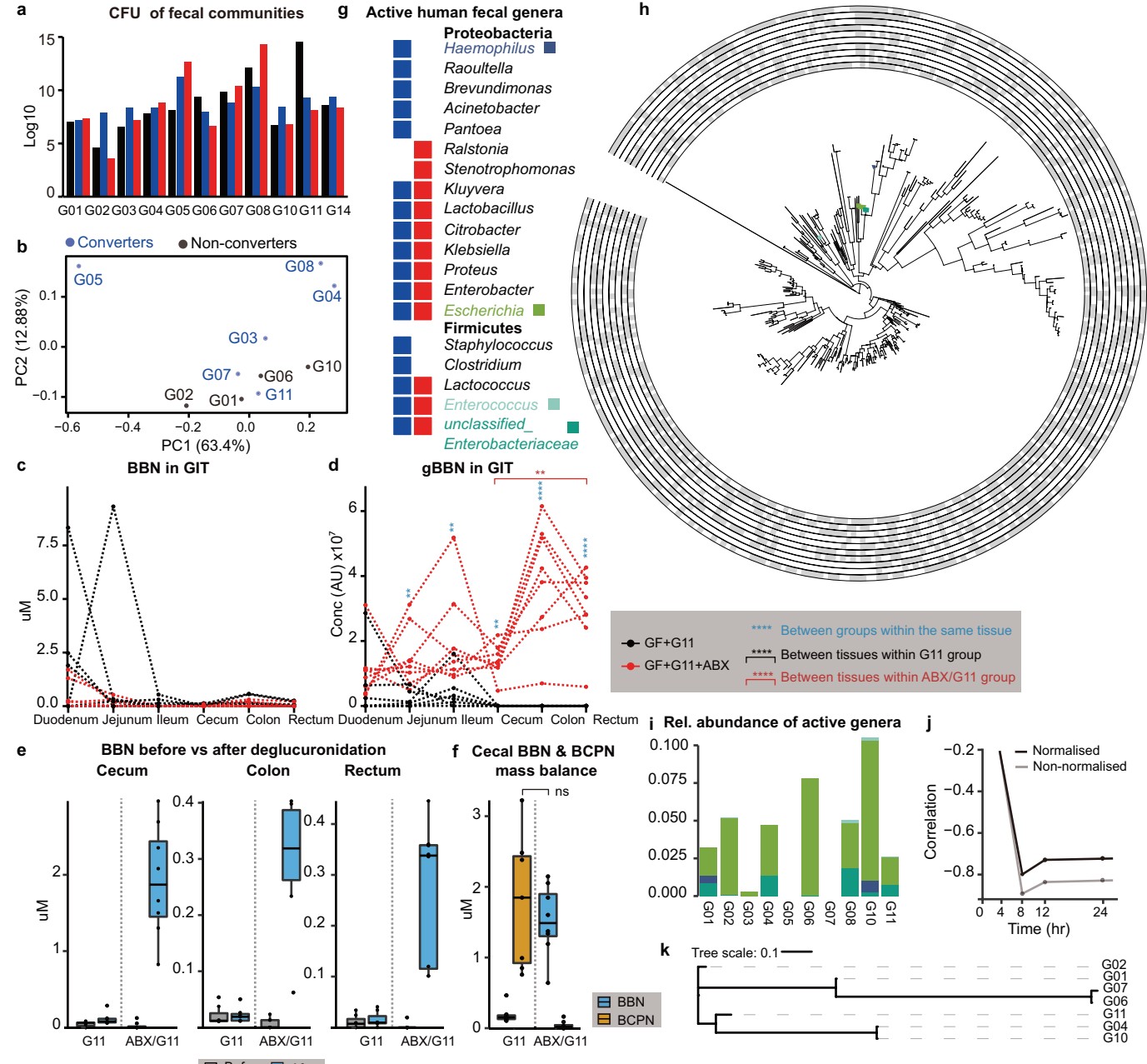

**Extended Data Fig. 4 | BBN-metabolizing bacterial species in the human gut microbiome. a**. Colony forming units (CFU) of ex vivo cultures of fecal samples collected from human subjects after 24 h of incubation under anaerobic (black), microaerobic (blue), and aerobic (red) conditions (Table S24). **b**. Principle component analysis (PCA) of fecal communities characterized with 16S rRNA sequencing of fecal microbiomes in **a** (Table S25). **c-d**. BBN and gBBN concentration along the gastrointestinal tract (GIT) of germ-free (GF) mice inoculated with fecal microbiota from subject G11 with (red) or without (red) antibiotic (ABX) co-treatment followed by 10 days of BBN administration via drinking water (Table S28). Pairwise comparisons between groups for the same tissue and between tissues within the same group were performed using estimated marginal means (EMMs) after fitting the data with a linear mixed model and *p*-values were calculated using a two-sided t-test and corrected for multiple testing using FDR (Table S29). **e**. Quantification of BBN in cecum, colon and rectum before and after in vitro β-glucuronidase treatment (Table S27). **f**. Mass balancing between BBN and BCPN levels in cecum following

in vitro β-glucuronidase treatment (Table S30). Pairwise comparisons between BBN and BCPN in different groups were performed with EMMs and two-sided Tukey test (n = 8/7 ABX-treated/untreated mice, Table S31). **g**. Bacterial genera isolated from fecal communities in **a** that converted BBN to BCPN under microaerobic (blue) and/or aerobic (red) conditions. **h**. Mapping of the active genera identified in **g** in the same fecal microbiomes characterized by 16S rRNA amplicon sequencing. **i**. Relative abundances of the genera in the respective microbiomes. **j**. Correlation between relative abundances of active ASV and BCPN production in individual subjects at 4, 8, 12 and 24 h of measurements. **k**. Tree representing the clustering of the genomes of Escherichia isolates from 7 different human subjects labelled with * in Fig. 4d, based on the binary presence and absence of accessory genes. *p*-values: 0.0001 ≤ ****, 0.001 ≤ ***, 0.01 ≤ **, 0.05 ≤ *. For each box plot, the center line represents the median, the box limits represent the 25th and 75th percentile, and the whiskers show 1.5× the interquartile range.

**ECPN in non-GIT tissues**

**Extended Data Fig. 5 | ECPN concentrations in non-GIT tissues after long-term treatment are comparable with or without antibiotics treatment.**
N-ethyl-N-(3-carboxypropyl)nitrosamine (ECPN) concentrations in liver, kidney and plasm of conventional mice treated with N-ethyl-N-(4-hydroxybutyl)-nitrosamine (EHBN) alone (black) or together with antibiotics (red) for 8 weeks (n = 9/8) (Table S43). For each box plot, the center line represents the median, the box limits represent the 25th and 75th percentile, and the whiskers show 1.5× the interquartile range.

Janos Terzic MD PhD

# Reporting Summary

## Statistics

For all statistical analyses, confirm that the following items are present in the figure legend, table legend, main text, or Methods section.

| n/a | Confirmed | |
|---|---|---|
| ☐ | ☒ | The exact sample size (*n*) for each experimental group/condition, given as a discrete number and unit of measurement |
| ☐ | ☒ | A statement on whether measurements were taken from distinct samples or whether the same sample was measured repeatedly |
| ☐ | ☒ | The statistical test(s) used AND whether they are one- or two-sided <br> *Only common tests should be described solely by name; describe more complex techniques in the Methods section.* |
| ☒ | ☐ | A description of all covariates tested |
| ☐ | ☒ | A description of any assumptions or corrections, such as tests of normality and adjustment for multiple comparisons |
| ☐ | ☒ | A full description of the statistical parameters including central tendency (e.g. means) or other basic estimates (e.g. regression coefficient) AND variation (e.g. standard deviation) or associated estimates of uncertainty (e.g. confidence intervals) |
| ☐ | ☒ | For null hypothesis testing, the test statistic (e.g. *F*, *t*, *r*) with confidence intervals, effect sizes, degrees of freedom and *P* value noted <br> *Give P values as exact values whenever suitable.* |
| ☒ | ☐ | For Bayesian analysis, information on the choice of priors and Markov chain Monte Carlo settings |
| ☒ | ☐ | For hierarchical and complex designs, identification of the appropriate level for tests and full reporting of outcomes |
| ☒ | ☐ | Estimates of effect sizes (e.g. Cohen's *d*, Pearson's *r*), indicating how they were calculated |

*Our web collection on statistics for biologists contains articles on many of the points above.*

## Software and code

Policy information about availability of computer code

| | |
|---|---|
| Data collection | MassHunter Qualitative Analysis Software (Agilent Technologies, version 10.0) was used to acquire LC-MS data. NIS-Elements Viewer 4.600 was used to acquire histological images. |
| Data analysis | MassHunter Qualitative Analysis Software (Agilent Technologies, version 10.0) was used to determine retention time for all compounds to enable targeted analysis and quantification. Peak integration was carried out using MassHunter Quantitative Analysis Software (Agilent Technologies, version 10.0).The following open access software tools and packages were used for the analysis of 16s rRNA data: qiime 2020.8, FastQC v0.11.5, cutadapt 2.10, dada2, vsearch 2.7.0, mafft 7.471, fasttree 2.1.11, iTOL v 6.0. Statistical analysis was performed with R 4.0.0. The following open access software tools and packages were used for the analysis of whole genome sequences: FastQC v0.11.93, MultiQC 1.12, Trimmomatic 0.39, Spades v3.15.3, QUAST v5.0.2, kraken v2.0.74, gtdbtk v2.1.1, mTAGs v1.0.4, Prokka v1.13, Roary v3.7.0, Phandango v1.3.0, iTOL v6. |

For manuscripts utilizing custom algorithms or software that are central to the research but not yet described in published literature, software must be made available to editors and reviewers. We strongly encourage code deposition in a community repository (e.g. GitHub). See the Nature Portfolio guidelines for submitting code & software for further information.

## Data

Policy information about availability of data

All manuscripts must include a data availability statement. This statement should provide the following information, where applicable:
- Accession codes, unique identifiers, or web links for publicly available datasets
- A description of any restrictions on data availability
- For clinical datasets or third party data, please ensure that the statement adheres to our policy

Raw sequencing data have been deposited on the ENA server, with accession number PRJEB43561 and raw data of the LC-MS analysis have been deposited in the Metabolights public repository with the accession number MTBLS3581. The RefSeq dataset of curated 16S ribosomal RNA sequences (see 16S RefSeq Nucleotide sequence records) was used as reference database for species assignation of full length 16S sequences. All other data are available in the main text or the supplementary materials.

## Research involving human participants, their data, or biological material

Policy information about studies with human participants or human data. See also policy information about sex, gender (identity/presentation), and sexual orientation and race, ethnicity and racism.

| | |
|---|---|
| Reporting on sex and gender | Both sexes were included |
| Reporting on race, ethnicity, or other socially relevant groupings | All participants were of Croatian residency and were not asked for ethnicity. |
| Population characteristics | Six male patients (58,2 ± 18,0, 33 to 81 years old) and 6 females (63 ± 8,1, 56-79 years old) were recruited during July and August 2020 at Split University Hospital. Patients underwent colonoscopy and gastroscopy for gastrointestinal disturbances. Criteria for patient selection were no antibiotic usage one month before sampling and absence of tumors in analysed tissues. |
| Recruitment | Patients were recruited by clinicians that were not in any other way included in subsequent analysis which is minimizing influence of bias of the study. Potential self-selection is difficult to assess. Selection bias could be present due to having used faecal samples from population with gastrointestinal problems, but authors regard that this does not lessen the value of results obtained with patients' microbiota, as the claims that human microbiota can convert BBN are still valid. |
| Ethics oversight | The study was approved by the ethical committee of University Hospital of Split (Permit number: 2181-147-01/06/M.S.-20-4) and the University of Split School of Medicine (Permit number: Ur. br. 2181-198-03-04-20-00400) and all participants gave informed consent before participation in the study. |

Note that full information on the approval of the study protocol must also be provided in the manuscript.

# Field-specific reporting

Please select the one below that is the best fit for your research. If you are not sure, read the appropriate sections before making your selection.

[x] Life sciences    [ ] Behavioural & social sciences    [ ] Ecological, evolutionary & environmental sciences

For a reference copy of the document with all sections, see nature.com/documents/nr-reporting-summary-flat.pdf

# Life sciences study design

All studies must disclose on these points even when the disclosure is negative.

| | |
|---|---|
| Sample size | No sample size calculation was performed. Instead, the key experiment was repeated five times independently resulting in 30 or more animals per group (Fig. 1b and 1c). In other experiments, sample sizes were kept at ten or less to adhere to 3R standards, but remaining sufficient for statistical comparisons. |
| Data exclusions | No data was excluded from the analysis. |
| Replication | Five independent experiments using 20-week exposure BBN or BBN/ABX were performed with in total 30/32 mice in each group respectively. For EHBN experiment, one independent experiment was performed with 8 EHBN/ABX, 9 EHBN treated mice. DBN/PBN experiments using metabolic cages were performed once with 5 mice per group. |
| | BBN metabolic cage experiments were repeated twice independently following 3, 6 or 7, and 12 weeks exposure to BBN with 5 animals per group. All attempts at replication were successful. |
| Randomization | Animals were randomly assigned to study groups. Human and mouse microbiota samples were allocated to all experimental groups as paired samples; therefore, random allocation was irrelevant to this set of measurements. |

| Blinding | Blinded histological assessment of urinary bladder tissues was performed by researchers and trained pathologists. However, blinding was not performed in any of the other experiments, since in some cases a single researcher was responsible for sacrificing, collecting, and processing samples. When one researcher was collecting while another performed measurements blinding was not planned in advance and thus not conducted. For human and mice microbiota analysis, blinding was not relevant since aim was to identify BBN-converting isolates. |
|---|---|

# Reporting for specific materials, systems and methods

We require information from authors about some types of materials, experimental systems and methods used in many studies. Here, indicate whether each material, system or method listed is relevant to your study. If you are not sure if a list item applies to your research, read the appropriate section before selecting a response.

## Materials & experimental systems

| n/a | Involved in the study |
|---|---|
| ✘ | ☐ Antibodies |
| ☐ | ✘ Eukaryotic cell lines |
| ✘ | ☐ Palaeontology and archaeology |
| ☐ | ✘ Animals and other organisms |
| ✘ | ☐ Clinical data |
| ✘ | ☐ Dual use research of concern |
| ✘ | ☐ Plants |

## Methods

| n/a | Involved in the study |
|---|---|
| ✘ | ☐ ChIP-seq |
| ✘ | ☐ Flow cytometry |
| ✘ | ☐ MRI-based neuroimaging |

## Eukaryotic cell lines

Policy information about cell lines and Sex and Gender in Research

| Cell line source(s) | HEP-G2 (ACC-180) cells were obtained from DSMZ-German Collection of Microorganisms and Cell Cultures GmbH. |
|---|---|
| Authentication | Cell line was not authenticated. |
| Mycoplasma contamination | Mycoplasma contamination was tested by PCR reaction on regular basis and used cell line tested negative. |
| Commonly misidentified lines (See ICLAC register) | Used cell line was not listed as a misidentified in ICLAC registry. |

## Animals and other research organisms

Policy information about studies involving animals; ARRIVE guidelines recommended for reporting animal research, and Sex and Gender in Research

| Laboratory animals | Two-months-old mice c57BL/6J were used in this study. |
|---|---|
| Wild animals | The study did not involve wild animals. |
| Reporting on sex | Most experiments were performed on male mice (as bladder cancer is 4 times more common in human males), but in some of them female mice were included (as described in the Methods). |
| Field-collected samples | The study did not involve samples collected from the field. |
| Ethics oversight | All mouse experiments were approved by the local IACUC (The University of Split, School of Medicine, Animal Welfare Committee and EMBL IACUC) and the national regulating authorities (Republic of Croatia Ministry of Agriculture, Veterinary and Food Safety Directorate) (permit numbers 525-10/0255-14-4,525-10/0255-15-5, 525-10/0543-21-8, 525-09/566-22-2 and 21-002_HD_MZ). |

Note that full information on the approval of the study protocol must also be provided in the manuscript.

## Plants

Seed stocks

*Report on the source of all seed stocks or other plant material used. If applicable, state the seed stock centre and catalogue number. If plant specimens were collected from the field, describe the collection location, date and sampling procedures.*

Novel plant genotypes

*Describe the methods by which all novel plant genotypes were produced. This includes those generated by transgenic approaches, gene editing, chemical/radiation-based mutagenesis and hybridization. For transgenic lines, describe the transformation method, the number of independent lines analyzed and the generation upon which experiments were performed. For gene-edited lines, describe the editor used, the endogenous sequence targeted for editing, the targeting guide RNA sequence (if applicable) and how the editor was applied.*

Authentication

*Describe any authentication procedures for each seed stock used or novel genotype generated. Describe any experiments used to assess the effect of a mutation and, where applicable, how potential secondary effects (e.g. second site T-DNA insertions, mosiacism, off-target gene editing) were examined.*

