## [Peer Review File · Nature]

Manuscript Title: Gut microbiota carcinogen metabolism causes distal tissue tumors

Reviewer Comments & Author Rebuttals

Reviewer Reports on the Initial Version:

Referee #1 (Remarks to the Author):

Roje, Zhang, et al. present an intriguing albeit preliminary set of experiments assessing the potential contributions of bacterial metabolism of the carcinogenic nitrosamine BBN to bladder cancer. The initial phenotype is clear and quite promising, with a marked reduction in the disease phenotype in mice (Figs. 1b,c). Unfortunately, despite a lot of work and data generated, the rest of the paper fails to definitively implicate gut bacterial metabolism in this phenotype or to provide much mechanistic insight into either the host response to BBN and its metabolites or the genes and enzymes involved in BBN metabolism on the microbial side. Additionally, multiple issues in the data analysis and interpretation were noted, raising concerns about scientific rigor and reproducibility.

Major issues:

1. Lack of statistical rigor. The use of statistics is inconsistently applied, with multiple important panels (e.g. Fig. 1d, Figs. 2c,e,f, Fig. 3a, Fig. 4a,c, etc.) lacking any statistical tests. Key contrasts are unclear and appear to lack adjustments for multiple hypotheses; for example, Figs. 2b,d. Perhaps most concerning, the authors make important conclusions based upon non-significant results; for example, the urinary BCPN levels in Fig. 1e and cecal levels of BCPN in Extended Data Fig. 2e. Taken together, these issues raise the concern that these data have not been rigorously vetted prior to submission. I'd recommend reviewing all the main and extended data figures with an expert statistician and/or a collaborator with a background in bioinformatics.
2. Lack of consideration for alternative hypotheses. There are multiple logical steps missing in between a phenotype in antibiotic depleted mice and the much more in-depth in vitro work on bacterial metabolism of BBN. First, it is critical to test if the antibiotics themselves could have impacted the phenotype, which could be evaluated through another delivery route (i.e. IP or IV) for the antibiotics or the use of germ-free mice exposed to BBN in the drinking water. Secondly, it remains unclear if gut metabolism matters at all for the phenotype, which could be tested by IP or IV delivery of the BBN in SPF mice or supplementation of BCPN to antibiotic depleted mice. If a germ-free model is established, one could then test if live bacteria are necessary for the phenotype. This model would also permit the colonization with the BBN metabolizing strains isolated later in the paper to assess if they are sufficient to increase BBN sensitivity. Taken together, while the impact of antibiotics on the disease is clear, the degree to which the microbiome or more specifically gut bacterial metabolism are responsible remains to be shown.

3. Limited biological replicates or repeat experiments. Replicate experiments are needed for Figs. 1 and 2. Even more concerning, the one case where a replicate experiment is shown (Extended Data Figs. 2c-e) does not report a significant difference, failing to replicate the prior experiment.

4. Ex vivo and isolate screening data doesn't address a central question. I appreciated the hard work for these large screens but didn't quite understand why they are in the current paper. The authors should try to integrate these data better with the original data in the mouse model; for example, by testing if the active species were depleted by the antibiotic regimen and/or adding back these active strains to the mouse model. More detail is also needed to interpret the data from the isolate screens, including sequencing results for all strains in the collection. It's just as important to note which strains aren't active as it is to say which ones are active.

5. Data on other body sites is distracting and complicates the story. The bulk of the paper is focused on gut bacterial metabolism, but pivots in the final section (Fig. 4) to now consider the lung microbiota. This raised questions as to the central hypothesis and whether lung bacterial metabolism is more relevant to the current model of disease. This data also raised concerns about the use of drinking water delivery of BBN and if that is relevant to the more common exposure through tobacco smoke. It now seems important to test an inhalation model, if available, for sensitivity to antibiotic depletion.

6. Bacterial oxidation is confounded by deconjugation. The importance of gut bacteria for deconjugation is well established and should be expected a priori. Given that BCPN could theoretically come from oxidation or deconjugation the relative importance of these two reactions is very difficult to disentangle. More definitive data is needed to support the role of oxidation, perhaps by cecal injection of BBN or by testing colonization with different oxidating strains in the presence or absence of E. coli or other beta-glucuronidase expressing bacteria. If oxidation was better supported by data, the impact of this paper would increase dramatically.

7. Extrapolating metabolic activity based only on genus and species level taxonomy is highly problematic. There are tons of examples of strain-level variation in xenobiotic metabolism in the literature. As such, the analyses presented in ED Figs. 3,4 are very difficult to interpret and hypothesis-generating in nature.

Additional points to address in revision:

Title: I'm not sure the bladder is really "distant" from the gut

Line 62: please include graphs for the alpha diversity data

Line 65: please add the PERMANOVA stats to the relevant figures

Line 85: cannot comment on effect size if the result is non-significant

Line 87: urinary BCPN was not significant

Line 97: claim about re-absorption is reaching and not well supported with the current data

Line 103: cecal results are not significant

Line 107: how was the trend, "tend to", identified? One group will always be higher than the other.

Line 137: 10% O₂ seems high for microaerobic – it would be interesting to test a full concentration gradient to see how much O₂ is required.

Line 143: the colon is not fully anaerobic

Lines 146, 181, 214: it's unclear how the isolates were generated and if they are fully axenic. It's also unclear how many non-redundant strains were tested, how diverse these collections are, or if they are representative of the human or mouse gut microbiota.

Line 154: terminology issue here and elsewhere, 16S is not metagenomics. Metagenomics refers to shotgun sequencing.

Lines 242-243: this seems very hard to believe – there's really no prior literature in this area? More importantly, this data falls short of implicating microbial metabolism so this should be restated to "antibiotic depletion impacts carcinogenesis and is associated with metabolite levels" or something to that effect.

Lines 276-277: the use of 5 drugs at high concentrations makes me even more worried that this is a drug-drug interaction not a drug-microbiome interaction.

Line 343: not metagenomics

Fig. 1a – include dose and specify which antibiotics were used

Fig. 1d,e – add stats to all panels

Figs. 2b, ED Fig. 2b – it's unclear what groups are being contrasted. Should compare between treatment groups at each location, not different locations within a group. It's also unclear how the timing of sacrifice could skew these results towards the cecum. And given the massive differences in levels between sites the data would be clearer on a log scale or as separate panels.

Fig. 2e,f – add stats

Fig. 2g – this is useful in framing the hypothesis but seems highly speculative given the current data

ED Fig. 1a – missing the no ABX control group at both timepoints. Need more replicates and stats.

ED Fig. 1b – add stats to this panel. % variation is super high on the x axis raising concerns about a data processing issue – recommend double-checking the code.

ED Fig. 1c – I'd opt to remove the urine microbiome since it is not a focus of this paper and raises questions as to contamination and sequencing methodology.

ED Fig. 2e – this is a non-significant result, need to discuss it accordingly

ED Fig. 2f – missing stats

ED Fig. 3a – add stats

ED Figs. 3b, 4c – very strange taxa isolated here – are these relevant in vivo?

ED Figs. 3c, 4d – what does this show?

Referee #2 (Remarks to the Author):

The authors present an excellently designed study demonstrating a clear link between metabolism of certain substances (nitrosamines) to carcinogenic metabolites by the gut microbiome (model system: urothelial carcinogenesis/bladder cancer). Furthermore, the authors can clearly demonstrate that the carcinogenic potency of nitrosamines can be almost completely suppressed by eradication of certain microbial commensals. To further enhance their results, they clearly demonstrate that bacterial strains/communities who are capable to oxidize nitrosamines to carcinogenic metabolites exist in human individuals and can be found at various locations where carcinogen induced tumors can arise (gut, upper aerodigestive tract, oral cavity). These novel results, which are also potentially highly clinically relevant, improve our knowledge of the influence of the microbiome on the chemical carcinogenesis of certain tumors. Results are overall well presented and conclusions are balanced and fully supported by the presented data. Where necessary or appropriate, relevant past literature has been cited. The manuscript has a clear ductus, is well written, and is easy to read and understand supported by the accurate illustrations. The present study is therefore of great relevance and also of great interest to the entire cancer research community. The experimental setting is well conducted and state of the art. Where applicable appropriate statistical test have been chosen for illustrating significant findings.

Therefore, from the reviewer's point of view, the publication of the present study can be fully endorsed as soon as the points listed below have been sufficiently addressed.

Major points:

- Although the BBN model is commonly used to study urothelial carcinogenesis, it represents a highly artificial system that does not represent the regular incorporation of urothelial carcinogens. For many chemical carcinogenesis sequences, it is now known that there is not a single relevant carcinogen, but rather multiple chemical compounds contributing to carcinogenesis (e.g., composition of cigarette smoke). Therefore, consideration should be given to validating the present results on a more realistic murine carcinogenesis model (e.g., cigarette smoke exposure model) that reflects the full range of the major risk factor of urothelial carcinoma (smoking). On the one hand, this would allow to illuminate and validate the actual influence of the microbiome (and indirectly the impact of nitrosamines in cigarette smoke induced carcinogenesis), but on the other hand, it would also significantly increase the translational and clinical relevance of the present studies, and additionally provide interesting results whether the microbiome significantly influences the carcinogenesis sequence even in the case of a polychemical carcinogen (cigarette smoke). The additional cigarette smoke model would further provide additional benefits and interesting aspects: The authors also studied the microbiome of the upper aerodigestive/gastrointestinal tract and hypothesize that flora alterations may have significant influence on carcinogenesis in these niches as well. Under certain conditions, a cigarette smoke model also allows the study of other cigarette smoke-induced carcinomas, especially
- squamous cell carcinomas of the head and neck including esophageal carcinomas

- lung carcinomas

Like urothelial carcinoma, these tumors classically arise via a chemical-toxic metaplasia/dysplasia carcinogenesis sequence. Therefore, a further step could be to investigate to what extent these sequences are also influenced by the microbiome. This would further massively increase the clinical/medical relevance.

- The second major point belongs to the issue of the BBN model. Although the authors clearly show that microbial strains from certain body locations (upper aerodigestive tract, lung, gut) are shared within the same human individual and are able to oxidize BBN to BCPN, it is not proven, that the strains which are found in human isolates are able to propagate the same cancerogenic effects as in mice. Thus, the authors should consider to proof the cancerogenic potential of human isolates in an experimental system, e.g. via isolation of specific BBN->BCPN converting strains, decontamination of murine gut microbiome and transplantation of human strains ◊ followed by BBN ingestion with subsequent ABX versus no ABX treatment ◊ analysis of murine bladders. (and if applied same procedure in tobacco smoke model!). This would significantly strengthen the biological rationale of the present study.

Minor points:

- Could any differences be observed in in caecum/colon mucosa between ABX/BBN vs. BBN groups? High intake of certain nitrosamine compounds (via red processed meat) are known to associate with an increased colorectal adenocarcinoma risk. It would be interesting to see if high levels of BCPN in the caecum might associate with precursors of colorectal carcinogenesis or changes in colon mucosa.

- Figure 1B: Dysplasia and CIS panels are not showing convincing cellular atypia. The dysplasia panel is rather showing a squamous metaplasia than a true urothelial dysplasia. It is known that BBN treated mice often develop squamous metaplasia. The image shown as CIS still shows a preserved luminal and basal polarity and there are no remarkable high grade atypia qualifying for CIS. From a pathological point of view there is no need to distinguish between early invasion and invasion. However, the panel with early invasion is not showing a convincing stroma invasion. Thus panels have to be updated with representative images showing dysplasia and CIS. Please remove the category of early invasion. Histopathological analysis could be further strengthened by objective correlates of neoplastic behavior, such as detection of specific oncogenic mutations (TP53 mutations, etc.).

Referee #3 (Remarks to the Author):

The data presented here are potentially interesting but preliminary. The finding that antibiotic treatment decreases the severity of BBN-induced bladder cancer is sound. From that point on, there are several open questions:

- Fig 1d shows a correlation between the level of BCPN level and tumorigenesis, but this doesn't prove that bladder cancer is driven by BCPN. I realize that prior studies have shown that BCPN treatment can induce cancer, but it is possible, in principle, that some other unknown metabolite of BBN is the primary driver in this model.

- The data in Fig 2 showing that BCPN is produced in the intestine are clear, but efforts to identify the organisms responsible are confusing. First, the requirement for O₂ is perplexing since the gut is anaerobic. Even if communities from other body sites are involved, the authors see the highest level of BCPN in the cecum (and much lower levels elsewhere). This is inconsistent with the requirement for oxygen.

- Almost certainly, the identification of *Corynebacterium* and *Staphylococcus* species that can catalyze the conversion of BBN to BCPN aerobically is not biologically relevant. Those organisms are skin colonists and present at very low levels in the gut.

- I think I understand what the authors are getting at in Fig 4c - they are trying to show that other communities might be involved. But doesn't the data from Fig 2 suggest that most of the BCPN is in the cecum? I am confused about whether enough BCPN is produced in other body sites to make a meaningful contribution to the cancer-inducing pool (again, if BCPN is really driving tumorigenesis in this model). It seems incomplete that it is left unclear which body site is relevant for conversion.

Most importantly, in addition to addressing the points above, I think the authors would need to show definitively, using germ-free mice colonized by defined communities +/- a BBN-to-BCPN converting strain (from Fig 3 or 4), that the presence of that strain alone is sufficient to potentiate BBN-induced tumorigenesis. Otherwise the study seems correlative but not definitive; it would be unclear whether the data from Figs 2-4 explain the mechanism of the initial observation.

Author Rebuttals to Initial Comments:

Referees' comments:

Referee #1 (Remarks to the Author):

Roje, Zhang, et al. present an intriguing albeit preliminary set of experiments assessing the potential contributions of bacterial metabolism of the carcinogenic nitrosamine BBN to bladder cancer. The initial phenotype is clear and quite promising, with a marked reduction in the disease phenotype in mice (Figs. 1b,c). Unfortunately, despite a lot of work and data generated, the rest of the paper fails to definitively implicate gut bacterial metabolism in this phenotype or to provide much mechanistic insight into either the host response to BBN and its metabolites or the genes and enzymes involved in BBN metabolism on the microbial side. Additionally, multiple issues in the data analysis and interpretation were noted, raising concerns about scientific rigor and reproducibility.

We thank the reviewer for their assessment of our study. Based on their comments, we have performed additional experiments and analyses to directly link microbial carcinogen metabolism to the observed microbiome-dependent tumorigenesis in bladder tissue. Further, we have repeated critical experiments to demonstrate and statistically test the reproducibility of observed effects.

Major issues:

1. Lack of statistical rigor. The use of statistics is inconsistently applied, with multiple important panels (e.g. Fig. 1d, Figs. 2c,e,f, Fig. 3a, Fig. 4a,c, etc.) lacking any statistical tests. Key contrasts are unclear and appear to lack adjustments for multiple hypotheses; for example, Figs. 2b,d. Perhaps most concerningly, the authors make important conclusions based upon non-significant results; for example, the urinary BCPN levels in Fig. 1e and cecal levels of BCPN in Extended Data Fig. 2e. Taken together, these issues raise the concern that these data have not been rigorously vetted prior to submission. I'd recommend reviewing all the main and extended data figures with an expert statistician and/or a collaborator with a background in bioinformatics.

We thank the reviewer for pointing out the inconsistent use of statistical test across the different data sets and figure panels. We have corrected this in the revised manuscript by explicitly indicating statistical tests used and by providing exact p-values in the figures. Further, we have undertaken new experiments to increase the sample size for all critical experiments. Specifically, to better elucidate the causative association between BCPN accumulation and bladder cancer development, we have repeated and combined data from different timepoint along the long-term BBN treatment (*i.e.*, 3, 6, 7 and 12 weeks) in Fig. 1d-e. These data corroborate that BCPN, the main metabolite of BBN, but not BBN itself significantly accumulates in both urinary bladder and urine in animals without antibiotic treatment.

Following the reviewer's suggestion, we also consolidated our statistical testing when comparing BCPN levels along the GIT tract. We first used linear mixed models to describe the relationship between BCPN levels and the groups and the tissues. Moreover, we corrected p-values to adjust for multiple testing using false discovery rate (FDR) and reported corrected p-values. Pairwise comparisons between groups for the same tissue and between tissues within the same group were done using estimated marginal means (EMMs) and p-values were adjusted with FDR (see also Supplementary Table S9 and S14). Key comparisons between

treatment groups and between intestinal locations within treatment groups are indicated in the revised Fig. 2b-d and Extended Data Fig. 2b-c, To facilitate figure reading we applied the following colour code: turquoise: testing between treatment groups; red: testing between intestinal sections of ABX/BBN-treated group; black: testing between intestinal sections of BBN-treated group.

Altogether, the performed statistical tests support our claims that i) caecal BCPN levels in BBN-treated mice are significantly higher than in ABX/BBN-treated animals (adj p-value: $3.39E-06$), ii) BCPN levels significantly increase in BBN-treated animals between ileum and cecum (adj p-value: $8.33E-04$), and iii) that BCPN produced in caecum is likely reabsorbed in the large intestine as levels between caecum and rectum significantly decrease (adj p-value: $4.36E-02$). This observation was reproduced when we inoculated germ-free mice with faecal contents from subject G11, which was shown to be metabolically active for BBN-to-BCPN conversion (Fig. 4a). Again, we observed that i) caecal, colon and rectum BCPN levels in inoculated mice are significantly higher than in germ-free animals (adj p-value: $6.89E-04$, $2.51E-09$ and $2.91E-02$, respectively), ii) BCPN levels significantly increase in inoculated animals between ileum and colon (adj p-value: $1.17E-05$), and iii) that BCPN produced in caecum and colon is likely reabsorbed as levels between colon and rectum significantly decrease (adj p-value: $1.22E-04$) (Fig 4b and Extended Data Fig. 4c-d).

2. Lack of consideration for alternative hypotheses. There are multiple logical steps missing in between a phenotype in antibiotic depleted mice and the much more in-depth in vitro work on bacterial metabolism of BBN. First, it is critical to test if the antibiotics themselves could have impacted the phenotype, which could be evaluated through another delivery route (i.e. IP or IV) for the antibiotics or the use of germ-free mice exposed to BBN in the drinking water. Secondly, it remains unclear if gut metabolism matters at all for the phenotype, which could be tested by IP or IV delivery of the BBN in SPF mice or supplementation of BCPN to antibiotic depleted mice. If a germ-free model is established, one could then test if live bacteria are necessary for the phenotype. This model would also permit the colonization with the BBN metabolizing strains isolated later in the paper to assess if they are sufficient to increase BBN sensitivity. Taken together, while the impact of antibiotics on the disease is clear, the degree to which the microbiome or more specifically gut bacterial metabolism are responsible remains to be shown.

We thank the reviewer for proposing additional experiments to further corroborate the causal link between gut bacterial BBN metabolism and bladder cancer. Following the reviewer's suggestion, we performed the following experiments:

- i) To test the effect of antibiotics on host metabolism of BBN, we treated germ-free mice with BBN in drinking water, with or without antibiotic treatment for 10 days to measure steady-state level of BCPN in bladder, liver, kidney, plasma and caecum (Extended Data Fig. 2d). These measurements showed no difference in BCPN levels between BBN-treated and ABX/BBN-treated animals in the absence of a microbiome (i.e. germ-free animals). We have added the description of this control experiment in the main text (line 94-98) and showed key results in Extended Data Fig. 2d and Table S15.
- ii) To test the direct impact of antibiotics on host BBN metabolism, we measured *ex vivo* BBN to BCPN conversion in liver HEPG2 cells in the presence and absence of antibiotic treatment. These results further demonstrates that the antibiotics do not interfere with host enzymes converting BBN to BCPN (Extended Data Fig. 2e). We have added the description of this control experiment in the main text (lines 98-103), show results in Extended Data Fig. 2e and Table S16.
- iii) To causally link bacterial BBN metabolism to altered BCPN toxicokinetics we established two different gnotobiotic mouse models:

First, we mono-colonized germ-free mice with BBN-metabolizing *E. flexneri* that we isolated from human faecal samples and characterized by genome sequencing and assembly (Fig. 4.d, Extended Fig. 4k, Table S34-35). Comparison of the intestinal levels of BBN and its metabolite between these mono-colonized and germ-free animals demonstrated that *E. flexneri* is also capable to metabolize BBN *in vivo* (Fig. 4f-g, and Table S37-38), reproducing the metabolic phenotype observed for animals colonized with a complex murine (Fig. 2b) and human gut community (Fig. 4b). We have added the description of this experiment in the main text (lines 235-240), show results in Fig. 4f-g, and Table S37-38, and describe the establishment of gnotobiotic mouse models in the material and methods part (line 314-347).

Second, we colonized germ-free animals with a synthetic bacterial community of either two gut bacterial type strains that do not metabolize BBN *in vitro* (Fig. 4c) or the same two strains and BBN-metabolizing *E. flexneri* in addition. Comparison of the intestinal levels of BBN and its metabolite between animals colonized with either this 2-membered or the 3-membered gut community demonstrated that *E. flexneri* is also capable to metabolize BBN *in vivo* as part of a defined bacterial gut community impacting *in vivo* BBN metabolism. We have added the description of this experiment in the main text (lines 240-247), show results in Fig. 4h-j, and Table S39-40, and describe the establishment of gnotobiotic mouse models in the material and methods part (line 314-347).

Altogether, these additional experiments demonstrate that the used antibiotics do not impact BBN metabolism of the host. Further, the establishment of gnotobiotic mouse models directly link the *in vivo* BBN metabolism to specific gut bacteria, isolated from human faecal communities. Hence, these results strongly strengthen the proposed link between the microbiota and altered *in vivo* BBN metabolism resulting in a higher incidence of bladder cancer.

3. Limited biological replicates or repeat experiments. Replicate experiments are needed for Figs. 1 and 2. Even more concerningly, the one case where a replicate experiment is shown (Extended Data Figs. 2c-e) does not report a significant difference, failing to replicate the prior experiment.

We do agree with the reviewer that the number of animals used for the BBN/BCPN *in vivo* and metabolic cage measurements shown in Fig. 1 and Fig. 2 were too limited for proper statistical analysis (also see comment 1 of reviewer 1). To address this limitation, we have performed additional mouse experiments to increase the number of replicates in both BBN-treated and ABX/BBN-treated animals. To take into account the long BBN-treatment (12 weeks), we collected samples at different time points (at 3, 6, 7, and 12 weeks). The repetition of the previous experiment and samples from different time points during BBN treatment consolidated our previous observation that antibiotic depletion reduces the intestinal BCPN production, urinary BCPN elimination and BCPN exposure of bladder tissue (revised Fig. 1 and Fig. 2). Measurement values of additional replicates and p-values of the comparison between groups are depicted in the revised Fig. 1, Fig. 2, Extended Data Fig. 2 and listed in Supplementary Table S9-11 and S14.

Furthermore, we introduce three additional mouse models that further consolidate the observation of the observed microbiota-dependent changes in BBN toxicokinetics: i) germ-free animals colonized with a human a fecal community and treated with BBN or ABX/BBN (Fig. 4b, Extended Data Fig. 4c-f and Table S26-31); ii) Comparison of the *in vivo* fate of BBN between germ-free mice and germ-free mice that were mono-colonized with BBN-metabolizing *E. flexneri* (Fig. 4f-g, and Table S37-38); iii) Comparison of the *in vivo* fate of BBN between germ-free mice colonized with either a BBN-metabolizing or BBN-non-metabolizing bacterial gut community (Fig. 4h-j, and Table S39-40).

We apologize for the apparent misunderstanding with respect to Fig. 2c-e in the original manuscript (now Extended Data Fig. 2i-l) and we would like to clarify the shown experiment is not a replication of the experiments shown in Fig. 2. *In vivo* data for BBN and BCPN discussed in the two paragraphs above stem from long-term BBN treatment experiments (BBN administered in drinking water for 10 days to 12 weeks) and hence show pseudo-steady state levels of BBN and BCPN. In contrast, the results shown in Fig. 2i-l stem from a single oral bolus administration of BBN to the animals and tissue analysis over the subsequent 9 hours. Based on our previous pharmacokinetic experiments with medical drugs (Zimmermann et al. *Science* 2019¹), this experimental design allowed us to test BCPN production independent of enterohepatic circulation and hence deglucuronidation activity (also see comment 6 of reviewer 1 for further explanations).

4. *Ex vivo* and isolate screening data doesn't address a central question. I appreciated the hard work for these large screens but didn't quite understand why they are in the current paper. The authors should try to integrate these data better with the original data in the mouse model; for example, by testing if the active species were depleted by the antibiotic regimen and/or adding back these active strains to the mouse model. More detail is also needed to interpret the data from the isolate screens, including sequencing results for all strains in the collection. It's just as important to note which strains aren't active as it is to say which ones are active.

We disagree with the reviewer in that the performed *ex vivo* microbiota BBN-conversion experiments do not contribute to the overall study conclusions. Having established that the BCPN is produced in the intestine in a microbiome dependent manner, the next logical step was to test whether the microbiota itself has the metabolic capacity to convert BBN to BCPN. Indeed, we could demonstrate that microbial *ex vivo* cultures from the human gut can metabolize BBN to BCPN in the presence of molecular oxygen (under aerobic and microaerobic conditions). With the goal to test whether such BBN-metabolizing activity is specific to the mouse gut microbiome or can be translated to the human gut microbiome, we performed the same *ex vivo* assays on human faecal cultures as well. As a result, we found that also human gut microbiota can produce BCPN from BBN. Furthermore, these experiments demonstrated that there are likely large interpersonal differences in gut microbial BBN metabolism. We believe that each of these three conclusions stemming from *ex vivo* BBN conversion assays with microbiota cultures strongly contribute to the overall conclusion of the study – namely i) that the mouse gut microbiota can metabolize organic carcinogens, ii) that these findings are translatable from the mouse model to the human gut microbiota, and iii) that gut microbial carcinogen metabolism may likely vary between individuals.

We however agree with the reviewer in that the BBN-metabolizing and non-metabolizing bacterial isolates could be better characterized and that these *in vitro* results could be better integrated into the context of the *in vivo* observations of the manuscript. To these aims, we have performed the following four additional experiments:

- i) Unfortunately, whole genome sequencing of all 42 BBN-metabolizing and many more non-metabolizing isolates (12 murine and 30 human isolates) would be beyond the scope of the current study and hence we restricted ourselves to perform systematic Sanger sequencing of the full-length 16S rRNA gene for the identification of these isolates. Additionally, to better characterize BBN-metabolizing bacteria isolated from human gut microbiota cultures, we focused on 7 strains of the same *Escherichia* species (based on full-length 16S rRNA sequencing) isolated from 7 different individuals. We sequenced and assembled the genome of these 7 isolates, demonstrating that they are distinct strains of the same bacterial species *Escherichia flexneri* (Extended Data Fig. 4k and Table S34-35).
- ii) As we isolated many *Escherichia* strains and strains from closely related species (*e.g.* *Escherichia fergusonii*, *Klebsiella* sp., *Citrobacter* sp.), we further tested how common BBN-conversion occurs

among a set of 95 well-characterized *Escherichia coli* strains. Strikingly, we found that all tested strains were indeed able to oxidize BBN to BCPN (Fig. 4e and Table S36). These results together with the large number of isolates closely related to *Escherichia sp.* suggest that BBN-metabolism is a common trait among different *Escherichia* species and strains.

- iii) Our mass spectrometry-based pipeline to specifically isolate active BBN-metabolizing bacteria, precludes us to characterize non-active bacterial isolates. To identify non-BBN-metabolizing gut bacteria as suggested by the reviewer, we tested a representative set of well-characterized gut bacterial type strains, which span the five most abundant phyla in the human gut (i.e. Bacteroidetes, Firmicutes, Actinobacteria, Proteobacteria and Verrucomicrobia) for their capacity to convert BBN to BCPN. Out of the tested 16 type strains, only *Escherichia coli* transformed BBN to BCPN (Fig. 4c and Table S32). These results are consistent with the BBN-metabolizing strains isolated from complex human gut communities. Further, this experiment resulted in a representative set of human gut bacteria that are not capable to convert BBN to BCPN. The results of this experiment are shown in Fig. 4c and Table S32 and described in the main text (line 194-198).
- iv) To directly link the *in vitro* work on gut bacterial BBN metabolism to the *in vivo* observation of gut bacterial contribution to BBN toxicokinetics, we established three different gnotobiotic mouse models based on the *in vitro* data: i) To translate the *ex vivo* data on BBN metabolism of human gut communities to the *in vivo* situation, we colonized germ-free mice with human faecal samples and repeated the BBN exposure experiment with a BBN and a BBN/ABX group. This experiment demonstrated that the microbiome-dependent changes of BBN toxicokinetics is not specific to the murine microbiome and can also be observed for mice carrying a 'humanized' gut microbiota (Fig. 4b, Extended Data Fig. 4c-f, and Table S26-31). ii) To demonstrate that *E. flexneri*, which we isolated from human stool samples due to its *in vitro* BBN-metabolism, also oxidizes BBN in the intestine, we mono-colonized germ-free animals with this strain prior to BBN treatment (Fig. 4f-g and Table S37-38). iii) We used the results of the screen of type strains for BBN metabolism, to assemble a BBN-metabolizing and non-metabolizing bacterial community for the colonization of germ-free mice and subsequent testing of *in vivo* BBN metabolism. These experiments further demonstrate that bacterial isolates that metabolize BBN *in vitro*, also contribute to BBN metabolism *in vivo* (Fig. 4h-j and Table S39-40).

In order to better integrate the results of the isolation effort to the gut microbiome data we matched BBN-metabolizing bacterial isolated (based on full length 16S rRNA sequencing) with ASVs (short-read 16S rRNA profiling) and quantified their abundance in murine caecum, colon, and rectum, as well as in the 10 human stool samples. This analysis demonstrated that BBN-metabolizing bacteria are low abundance in both the murine (<3%) and the human (<1%) gut microbiome. These analyses are shown in Fig. 3b-c and Extended Data Fig. 4g-i.

Given the staggering reduction of the overall bacterial load upon ABX treatment (CFU 99.99% lower in ABX-treated versus non-treated animals, as reported in Line 60, Extended Data Fig. 1a and Table S1) we strongly believe that the observed reduction of cancer incidence in ABX is caused by a depletion of the bacterial load in the gut rather than compositional microbiome changes. Additionally, the compositional nature of 16S rRNA data together with the 99.99% reduction of bacterial load precludes any conclusion on linking the taxonomic microbiome composition with its *in vivo* BBN-metabolizing activity between the two mouse groups.

5. Data on other body sites is distracting and complicates the story. The bulk of the paper is focused on gut bacterial metabolism, but pivots in the final section (Fig. 4) to now consider the lung microbiota. This raised questions as to the central hypothesis and whether lung bacterial metabolism is more relevant to the

current model of disease. This data also raised concerns about the use of drinking water delivery of BBN and if that is relevant to the more common exposure through tobacco smoke. It now seems important to test an inhalation model, if available, for sensitivity to antibiotic depletion.

We do agree with the reviewer (and also reviewer 3) that the data on the different body sites are rather distracting. Therefore, we follow the reviewer's suggestion and excluded this part from the revised manuscript.

6. Bacterial oxidation is confounded by deconjugation. The importance of gut bacteria for deconjugation is well established and should be expected a priori. Given that BCPN could theoretically come from oxidation or deconjugation the relative importance of these two reactions is very difficult to disentangle. More definitive data is needed to support the role of oxidation, perhaps by cecal injection of BBN or by testing colonization with different oxidating strains in the presence or absence of *E. coli* or other beta-glucuronidase expressing bacteria. If oxidation was better supported by data, the impact of this paper would increase dramatically.

We agree with the reviewer that gut bacterial contribution to BBN toxicokinetics comprises of the combination of a deconjugation and an oxidation reaction, which are not simple to disentangle experimentally. To demonstrate that bacterial BBN oxidation, demonstrated *in vitro*, also plays a role *in vivo* we have performed the following experiments:

- i) We quantified the absolute amounts of caecal BBN, glucuronyl-BBN (gBBN), and BCPN in BBN-treated and ABX/BBN-treated animals to estimate the mass balance of these compounds in the gastrointestinal compartments between the two groups. This analysis demonstrates that i) *in vivo* deglucuronidation only occurred in the presence of microbiota as BBN was only released in ABX/BBN after post-experimental deglucuronidation (Fig. 2e, Table S8, S9), ii) the total amounts of BBN released in the GIT of ABX/BBN group after post-experimental deglucuronidation balances out the total BCPN amounts in the BBN-treatment group (Fig. 2f, Table S8, S10), iii) no additional BCPN was released after post-experimental deglucuronidation in either of the groups (Extended Data Fig. 2m, Table S8, S9). These findings are further corroborated by the mass balance estimates performed for the gnotobiotic mouse experiments described above, in which we compared BBN metabolism between germ-free mice and human-feces-colonized mice (Extended Data Fig. 4e-f and Table S30-31).
- ii) To separate gut bacterial BBN-deglucuronidation activity from BBN-oxidation *in vivo*, we designed a mouse experiment to follow the kinetics of BBN and its metabolites in the intestine after a single oral dose of BBN. The rationale for this experimental design is the fact that BBN absorption from the gut after oral administration, hepatic glucuronidation, and secretion of gBBN leads to a temporal delay of gBBN appearance in the gut while non-absorbed BBN is already present in the intestine. Hence, during the first hour after oral administration, bacterial BBN oxidation in the large intestine can be observed independent of deglucuronidation activity. Indeed, we found BCPN being produced in a microbiome-dependent manner (BBN versus ABX/BBN comparison) already one hour after oral administration and before we could detect any intestinal gBBN in the ABX/BBN control group (Extended Data Fig. 2i-l). These results hence demonstrate that bacterial BBN-oxidation to BCPN occurs in the large intestine of mice, also independently of BBN deglucuronidation.

7. Extrapolating metabolic activity based only on genus and species level taxonomy is highly problematic. There are tons of examples of strain-level variation in xenobiotic metabolism in the literature. As such, the analyses presented in ED Figs. 3,4 are very difficult to interpret and hypothesis-generating in nature.

We share the reviewer's concern for the general difficulty of functional extrapolation based on genus level taxonomy. However, the goal of the analysis presented in Extended Data Figs. 3 and 4 was not to extrapolate metabolic activity, but to filter BBN-oxidizing bacterial isolates for their *in vivo* relevance and to get an estimate of the abundance of potentially BBN-metabolizing genera in the gut microbiome:

- i) Since the protocol to isolate bacterial strains that metabolize BBN to BCPN included microaerobic enrichment, the identified strains could potentially be biased by this isolation procedure. Therefore, we filtered bacterial isolates based on their detectability in gut microbiome data (16S rRNA sequences) of the starting material (cecal, fecal and stool samples). To this aim, we sequenced the full-length 16S rRNA gene of the isolates and only report an isolate, if we could map it to the 16S rRNA amplicon sequencing data of the original community. We have further clarified this procedure in the main text (line 157 - 162) and materials and methods (line 421 – 434). Results are displayed in Extended Data Fig. 3c and 4h and raw data compiled in Table S22, S25.
- ii) To estimate the abundance of BBN converters in large intestinal microbiomes, we quantified the relative abundance of BBN-converting bacterial taxa in the 16S rRNA amplicon data of murine caecal, rectal and faecal, as well as human stool samples. To this aim, quantification was not based on matching taxonomy but rather on direct mapping of the ASV representative sequences to the BBN-converting 16S reference sequence using vsearch, which was in turn used to assign consensus taxonomy. Although quantification based on 16S rRNA amplicon sequencing is neither exact nor at the taxonomic resolution to take strain differences into account, this analysis suggested that BBN converting bacteria are generally at low abundance in the human gut. We have further clarified this analysis and its limitations due to amplicon sequencing (i.e. relative quantification and limited taxonomic resolution) in the main text (line 162 – 167 and 214 - 224) and materials and methods (line 421 – 434). Results are displayed in Fig. 3c and Extended Data Fig. 4i and raw data compiled in Table S22, S25.

To gain insights into strain-specific differences in BBN metabolism, we performed two complementary experiments:

- i) We sequenced the whole genome of seven *Escherichia* isolates from different individuals that we found to oxidize BBN to BCPN. Phylogenetic analysis confirmed that these 7 isolates are different strains of the same species. These results are described in the main text (line 225- 231) and materials and methods (line 436 – 453) and results are shown in Extended Data Fig. 4k and Table S34-35.
- ii) To test the strain-specificity of BBN metabolism at a broader scale, we tested 95 different *E. coli* strains, previously isolated and characterized, for their ability to convert BBN to BCPN. We found that all of them can convert BBN to BCPN in our *in vitro* assay. These results suggest that BBN metabolism is not very strain specific, at least for *Escherichia coli*. The results of this assay are described in the main text (line 231 - 235) and shown in Fig. 4e and Table S36.

Additional points to address in revision:

Title: I'm not sure the bladder is really "distant" from the gut

We do agree in that 'distal' would be a more appropriate term to describe physiological distance between gut and bladder. We have changed the title accordingly.

Line 62: please include graphs for the alpha diversity data

We included the graph of the alpha diversity data as Extended Data Fig. 1c.

Line 65: please add the PERMANOVA stats to the relevant figures

We have added PERMANOVA test statistics for the analysis of microbiome data and to demonstrate that the gut microbiome composition is perturbed by antibiotics (p-value 0.001, 999 permutations). Data are shown in Extended Data Fig. 1b.

Line 85: cannot comment on effect size if the result is non-significant

We thank the reviewer for this comment and have corrected the mistake. See also extended response to comment 1 about replicate measurements and statistical tests applied.

Line 87: urinary BCPN was not significant

Following additional replication of the metabolic cage experiment, urinary BCPN shows significantly increased levels and increased urinary BCPN elimination in BBN-treated versus ABX/BBN-treated animals (Fig. 1d-e). Also see response to comment 1.

Line 97: claim about re-absorption is reaching and not well supported with the current data

The data from the additional mouse experiments performed for this manuscript revision independently support our finding of large intestinal absorption of BCPN. These data demonstrate that the BCPN concentration significantly decreases from caecum to rectum in i) SPF mice (Fig. 2b and Extended Data Fig. 2b and Table S7, S9, S13-14), ii) germ-free mice colonized with human faecal material (Fig. 4b and Table S26, S29), and germ-free mice mono-colonized with *E. flexneri* (Fig. 4g and Table S37-38).

Line 103: cecal results are not significant

Following additional replication of the mouse experiment shows significantly increased BCPN levels in BBN-treated versus ABX/BBN-treated animals (Fig. 1b). Also see response to comment 1.

Line 107: how was the trend, "tend to", identified? One group will always be higher than the other.

We do agree with the reviewer that the difference in BBN levels between BBN and BBN/ABX-treated animals is not the most convincing argument for intestinal oxidation of BBN to BCPN. The newly performed mass balance estimates of BBN and its metabolites in the caecum, described above, suggests that the majority of

intestinal BCPN is produced by bacteria in a two-step reaction to first deglucuronidate gBBN and second oxidize the liberated BBN to BCPN. Therefore, we emphasize in the revised manuscript the mass balance of BBN and its metabolites, together with the fact that free BBN alone cannot explain the high levels of BCPN observed. We have changed the main text (line 104 – 115) and we have adjusted the scale of Fig. 2c (caecal BBN concentrations) for easier comparison with Fig. 2b (caecal BCPN concentration).

Line 137: 10% O₂ seems high for microaerobic – it would be interesting to test a full concentration gradient to see how much O₂ is required.

We do agree that it would be interesting to evaluate an entire range of oxygen concentration to identify the oxygen levels required for different bacteria to oxidize BBN to BCPN. However, we feel that this question is beyond the scope of the current manuscript and should be addressed in future studies.

Line 143: the colon is not fully anaerobic

We agree that the colon is not fully anaerobic, which we believe is enabling oxygen-dependent oxidation of BBN to BCPN. Therefore, we write in the revised manuscript that the large intestine is ‘mostly anaerobic’.

Lines 146, 181, 214: it’s unclear how the isolates were generated and if they are fully axenic. It’s also unclear how many non-redundant strains were tested, how diverse these collections are, or if they are representative of the human or mouse gut microbiota.

Indeed, the tested isolates were axenic. They were generated by picking from dilution-streaking plates, re-cultured in liquid media. Sanger sequencing of the full-length 16s rRNA gene was used to determine the identity of the isolates. In case of “unambiguous sequencing reads”, isolates were discarded and not included in the analysis. For most of the BBN-metabolizing bacteria we isolated several isolates with identical full-length 16s rRNA sequences, which indicates that our sampling strategy approached isolation saturation for BBN-metabolizers from a given community. The complete list of BBN-metabolizing bacterial isolated, including replicate isolates is now provided in Table S50 and S51.

As described above (see also response to comment 4), our isolation protocol did not identify bacterial isolates that do not metabolize BBN by full-length 16s rRNA sequencing. Therefore, we decided to also test 16 type strains representative for the diversity of the human microbiome, covering 5 different bacterial phyla. We found that only one of these strains, *E. coli*, was capable to metabolize BBN (Fig. 4c and Table S32). This well mirrors the results from our isolation efforts, from which *Escherichia* was one of the major bacterial genera found to metabolize BBN, in both murine and human gut communities (Fig. 3b and 4d).

Line 154: terminology issue here and elsewhere, 16S is not metagenomics. Metagenomics refers to shotgun sequencing.

We fully agree and thank the reviewer for pointing this out. We have corrected ‘metagenomics analysis’ by ‘16S rRNA sequencing’ or ‘taxonomic profiling’ throughout the text.

Lines 242-243: this seems very hard to believe – there’s really no prior literature in this area? More importantly, this data falls short of implicating microbial metabolism so this should be restated to “antibiotic depletion impacts carcinogenesis and is associated with metabolite levels” or something to that effect.

We have followed the reviewer's suggestion and rephrased this statement in the text to "In this study, we demonstrated that microbial biotransformation of environmental carcinogens is a contributing factor for cancer development."

Lines 276-277: the use of 5 drugs at high concentrations makes me even more worried that this is a drug-drug interaction not a drug-microbiome interaction.

Taking the reviewer's concern about the impact of antibiotics serious (also raised in comment 2), we have performed additional control experiments that demonstrated that antibiotics do not directly interfere with BBN metabolism and its toxicokinetics. In brief, we exposed germ-free mice to BBN or BBN/ABX to show that BBN and BBN metabolite levels were not different in these two groups in the absence of a microbiome (Extended Data Fig. 2d). Further, using HepG2 liver cells, we showed that conversion of BBN to BCPN is not impacted by the presence of antibiotics (Extended Data Fig. 2e).

Line 343: not metagenomics

Corrected.

Fig. 1a – include dose and specify which antibiotics were used

The antibiotics and their concentrations used in the experiments are described in material and methods (line 315 – 319).

Fig. 1d,e – add stats to all panels

Done. Also see response to comment 1.

Figs. 2b, ED Fig. 2b – it's unclear what groups are being contrasted. Should compare between treatment groups at each location, not different locations within a group. It's also unclear how the timing of sacrifice could skew these results towards the cecum. And given the massive differences in levels between sites the data would be clearer on a log scale or as separate panels.

To clarify the representation of intestinal data, we have introduced three colours of asterisks guiding the reader through the different comparisons performed: i) turquoise: comparisons of the same body compartment between different experimental groups; ii) black: comparisons of different body compartments within the same experimental groups, BBN-treated; iii) red: comparisons of different body compartments within the same experimental groups, BBN/ABX-treated.

To ensure that the timing of sacrifice does bias our results, mice from different groups were sacrificed interchangeably.

We decided to display the data not in log scale to emphasize the large accumulation of BCPN in the cecum.

Fig. 2e,f – add stats

Done.

Fig. 2g – this is useful in framing the hypothesis but seems highly speculative given the current data

We do agree with the reviewer and we removed this scheme from the figure.

ED Fig. 1a – missing the no ABX control group at both timepoints. Need more replicates and stats.

CFU were counted for after 0, 2, 3, 8 and 12 weeks of antibiotic treatment with more replicates added (Extended Data Fig. 1a and Table S1).

ED Fig. 1b – add stats to this panel. % variation is super high on the x axis raising concerns about a data processing issue – recommend double-checking the code.

We did double check the code and get the same high variation. However, such high variation between two groups, one untreated and the other treated with a cocktail of antibiotics, is actually little surprising given that the applied antibiotic treatment reduced the bacterial load by four orders of magnitude (99.99%) based on cfu counts.

ED Fig. 1c – I'd opt to remove the urine microbiome since it is not a focus of this paper and raises questions as to contamination and sequencing methodology.

We do agree with the reviewer and we removed information about the urine microbiome from the manuscript.

ED Fig. 2e – this is a non-significant result, need to discuss it accordingly

Please refer to the reply to comment 1.

ED Fig. 2f – missing stats

Has been added.

ED Fig. 3a – add stats

Has been added.

ED Figs. 3b, 4c – very strange taxa isolated here – are these relevant in vivo? We do agree with the reviewer in that some of bacteria isolated are not typical gut species. We think that these isolates likely got picked up in our isolation protocol due to the enrichment in the presence of oxygen (aerobic and microaerobic conditions), which likely enriched for very low abundant or even ingested bacteria. To filter out species that are not detectable in gut microbial communities, we mapped the full-length 16s rRNA gene onto the 16S rRNA amplicon microbiome data of the community, from which the respective isolate was derived. We then only kept isolates that we could map to the microbiome data for the revised Fig. 3b and 4c (now 4d). For human, the BBN-converting bacteria that we could map to the intestinal microbiome are indeed expected gut bacteria, such as *Escherichia sp.* and *Enterococcus sp.* (Fig. 4d). For mouse, numerous BBN-converting bacteria that we could map to the intestinal microbiome are rather skin-associated than gut associated, such as *Corynebacterium sp.* and *Staphylococcus sp.* (Fig. 4d). We think that the detectable levels of such skin associated bacteria in the murine gut microbiome is likely due to their extensive grooming activity. We describe the performed procedure in the main text (line 157 – 162 and line 209 -216) and materials and methods (line 421 – 434). An overview of all BBN-metabolizing isolates is provided in Table S21.

ED Figs. 3c, 4d – what does this show?

Please, see response to comment 7.

Referee #2 (Remarks to the Author):

The authors present an excellently designed study demonstrating a clear link between metabolism of certain substances (nitrosamines) to carcinogenic metabolites by the gut microbiome (model system: urothelial carcinogenesis/bladder cancer). Furthermore, the authors can clearly demonstrate that the carcinogenic potency of nitrosamines can be almost completely suppressed by eradication of certain microbial commensals. To further enhance their results, they clearly demonstrate that bacterial strains/communities who are capable to oxidize nitrosamines to carcinogenic metabolites exist in human individuals and can be found at various locations where carcinogen induced tumors can arise (gut, upper aerodigestive tract, oral cavity). These novel results, which are also potentially highly clinically relevant, improve our knowledge of the influence of the microbiome on the chemical carcinogenesis of certain tumors. Results are overall well presented and conclusions are balanced and fully supported by the presented data. Where necessary or appropriate, relevant past literature has been cited. The manuscript has a clear ductus, is well written, and is easy to read and understand supported by the accurate illustrations. The present study is therefore of great relevance and also of great interest to the entire cancer research community. The experimental setting is well conducted and state of the art. Where applicable appropriate statistical test have been chosen for illustrating significant findings. Therefore, from the reviewer's point of view, the publication of the present study can be fully endorsed as soon as the points listed below have been sufficiently addressed.

We thank the reviewer for the positive and supporting feedback on our study.

Major points:

- Although the BBN model is commonly used to study urothelial carcinogenesis, it represents a highly artificial system that does not represent the regular incorporation of urothelial carcinogens. For many chemical carcinogenesis sequences, it is now known that there is not a single relevant carcinogen, but rather multiple chemical compounds contributing to carcinogenesis (e.g., composition of cigarette smoke). Therefore, consideration should be given to validating the present results on a more realistic murine carcinogenesis model (e.g., cigarette smoke exposure model) that reflects the full range of the major risk factor of urothelial carcinoma (smoking). On the one hand, this would allow to illuminate and validate the actual influence of the microbiome (and indirectly the impact of nitrosamines in cigarette smoke induced carcinogenesis), but on the other hand, it would also significantly increase the translational and clinical relevance of the present studies, and additionally provide interesting results whether the microbiome significantly influences the carcinogenesis sequence even in the case of a polychemical carcinogen (cigarette smoke). The additional cigarette smoke model would further provide additional benefits and interesting aspects: The authors also studied the microbiome of the upper aerodigestive/gastrointestinal tract and hypothesize that flora alterations may have significant influence on carcinogenesis in these niches as well. Under certain conditions, a cigarette smoke model also allows the study of other cigarette smoke-induced carcinomas, especially
 - squamous cell carcinomas of the head and neck including esophageal carcinomas
 - lung carcinomas

Like urothelial carcinoma, these tumors classically arise via a chemical-toxic metaplasia/dysplasia carcinogenesis sequence. Therefore, a further step could be to investigate to what extent these sequences are also influenced by the microbiome. This would further massively increase the clinical/medical relevance.

We do agree with the reviewer in the potential translational implication of the current study. We also agree that the used BBN-induced urothelial cancer model does not fully reflect the multifactorial nature of (cigarette smoke-induced) urothelial carcinogenesis. However, the goal and main strength of the current

study, in our opinion, is the mechanistic dissection of gut microbial contribution to the toxicokinetics of carcinogens at the molecular level, rather than modelling the multifactorial effects that smoke plays in tumorigenesis. Therefore, we used the BBN-induced bladder cancer model, which is one of the best-established models to study chemical carcinogenesis, as the reviewer has also pointed out. As the carcinogenesis mechanisms (i.e. BBN to BCPN conversion and accumulation of BCPN in urothelial tissue) have been extensively investigated in previous studies²⁻⁵, we could build on this knowledge to directly quantify the impact of gut bacterial metabolism on these tumorigenic processes. In contrast, the large number of smoke-derived chemicals and additional smoke-related factors that lead to tumorigenesis would make it challenging to interpret results of a smoke-induced cancer model and the impact of the gut microbiota. Therefore, we decided not to establish a smoke-induced tumour model. Instead of the cigarette smoke-induced cancer model, we tested 3 additional nitrosamine compounds that have previously been established to model chemical induced tumorigenesis in mice. We could demonstrate that the microbiota impacts the toxicokinetics of these additional nitrosamine compounds the same way as it affects BBN toxicokinetics (Fig. 5a-d, and Table S41). Furthermore, we were able to demonstrate microbiota-dependent bladder tumorigenesis for one of the compounds (EHBN). Altogether, the experiments performed with these additional chemicals, together with the developed gnotobiotic mouse models (described above) both corroborate and generalize our findings. Following the reviewer's suggestion, we believe that future studies should include more complex cancer models in order to facilitate the translation to clinics.

- The second major point belongs to the issue of the BBN model. Although the authors clearly show that microbial strains from certain body locations (upper aerodigestive tract, lung, gut) are shared within the same human individual and are able to oxidize BBN to BCPN, it is not proven, that the strains which are found in human isolates are able to propagate the same cancerogenic effects as in mice. Thus, the authors should consider to proof the cancerogenic potential of human isolates in an experimental system, e.g. via isolation of specific BBN->BCPN converting strains, decontamination of murine gut microbiome and transplantation of human strains → followed by BBN ingestion with subsequent ABX versus no ABX treatment → analysis of murine bladders. (and if applied same procedure in tobacco smoke model!). This would significantly strengthen the biological rationale of the present study.

We appreciate the reviewer's comment reproducing the phenotype using human-derived strains and agree that it would be of great relevance to demonstrate their *in vivo* relevance in a mouse model. To this end, we established three different models using germ-free mice that we colonized with either a BBN-metabolizing human microbiota, a single human-derived, BBN-metabolizing bacterial strain, or two synthetic gut bacterial communities (one with BBN-metabolizing capacity and the other without). These experiments are described in details in the response to comment 4 of reviewer 1 and key results are shown in Fig. 4b and Fig. 4f-j, Extended Data Fig. 4c-f, and Table S28-31 and Table S37-40.

Together, these experiments directly link the *in vitro* work on gut bacterial BBN metabolism and the identified bacterial BBN-metabolizing bacterial isolates to the *in vivo* observation of gut bacterial contribution to BBN toxicokinetics. Additionally, the established gnotobiotic mouse models provide an excellent system for future studies aiming at investigating the impact of the gut microbiota on carcinogen metabolism.

Minor points:

- Could any differences be observed in in caecum/colon mucosa between ABX/BBN vs. BBN groups? High intake of certain nitrosamine compounds (via red processed meat) are known to associate with an increased colorectal adenocarcinoma risk. It would be interesting to see if high levels of BCPN in the caecum might associate with precursors of colorectal carcinogenesis or changes in colon mucosa.

We have not observe any intestinal tumorigenesis in either of the groups.

- Figure 1B: Dysplasia and CIS panels are not showing convincing cellular atypia. The dysplasia panel is rather showing a squamous metaplasia than a true urothelial dysplasia. It is known that BBN treated mice often develop squamous metaplasia. The image shown as CIS still shows a preserved luminal and basal polarity and there are no remarkable high grade atypia qualifying for CIS. From a pathological point of view there is no need to distinguish between early invasion and invasion. However, the panel with early invasion is not showing a convincing stroma invasion. Thus panels have to be updated with representative images showing dysplasia and CIS. Please remove the category of early invasion. Histopathological analysis could be further strengthened by objective correlates of neoplastic behavior, such as detection of specific oncogenic mutations (TP53 mutations, etc.).

We thank the reviewer proposing an improved tumour classification. We followed the reviewer's suggestion and removed the category of 'dysplasia', which is now part of the 'non-cancerous' group. With respect to the reviewer's comment on 'carcinoma in situ (CIS)', we would like to draw their attention on the epithelial changes that are associated with invasive tumours (please check the panel below). We have used these epithelial changes as standard for CIS classification in mice, even though these epithelial changes moderately differ from human CIS. Furthermore, we removed the 'early invasion category' from the tissue classification, because previously categorized 'early invasive' samples are now reclassified as CIS, and because they are not showing convincing stromal invasion, as pointed out by the reviewer. Importantly, our overall pathological findings are not affected by this re-classification and the difference between groups is still highly significant (Fig. 1c, p-value = 1.6E-05).

CIS associated with invasive tumours (Ep – epithelium, Tu – tumour):

Referee #3 (Remarks to the Author):

The data presented here are potentially interesting but preliminary. The finding that antibiotic treatment decreases the severity of BBN-induced bladder cancer is sound. From that point on, there are several open questions:

- Fig 1d shows a correlation between the level of BCPN level and tumorigenesis, but this doesn't prove that bladder cancer is driven by BCPN. I realize that prior studies have shown that BCPN treatment can induce cancer, but it is possible, in principle, that some other unknown metabolite of BBN is the primary driver in this model.

We appreciate the reviewer's thorough consideration for the alternative BBN metabolic pathways. Indeed, additional BBN metabolites have been described in literature, but it is well accepted in the field that BCPN and gBBN are the main metabolites in the body and that urothelial BBN and BCPN exposure triggers tumorigenesis^{2,3}. This was demonstrated by direct BCPN administration in drinking water, which reproduced the histopathological tumour phenotype as after equivalent BBN administration via either drinking water⁴ or intravesicular instillation⁵. Given that BCPN has been reproducibly observed to accumulate in the bladder in previous studies, our point was that the increased production of BCPN indicates an overall increase of carcinogenic substrates in the system for more DNA adduct formation. Although we cannot exclude that there are metabolites other than BCPN contributing to the observed microbiota-dependent cancer phenotype, we think that our data reproducibly suggest that bacterially-produced BCPN is an important contributor in the used BBN-induced bladder cancer model.

- The data in Fig 2 showing that BCPN is produced in the intestine are clear, but efforts to identify the organisms responsible are confusing. First, the requirement for O₂ is perplexing since the gut is anaerobic. Even if communities from other body sites are involved, the authors see the highest level of BCPN in the cecum (and much lower levels elsewhere). This is inconsistent with the requirement for oxygen.

We do agree with the reviewer (and also reviewer 1) in that the data on other body sites is rather confusing and does not contribute to the main conclusions of the manuscript. Therefore, we decided to exclude these data in the revised manuscript.

With respect to oxygen in the large intestine, we disagree with the reviewer in that the lower intestine is strictly anaerobic. Multiple studies have demonstrated that there is an oxygen gradient across the large intestine, due to the fact that oxygen from oxygenated host tissue diffuses into the intestinal lumen. Further, it is thought that microaerobic and facultative aerobic strains contribute to this gradient by utilizing the oxygen diffusing out of the gut epithelial tissue^{6,7}. Experimentally, we could demonstrate (Fig. 3a and 4a) that both murine and human gut microbial communities only convert BBN to BCPN in the presence of oxygen (aerobic and microaerobic conditions). Therefore, we used the same conditions to isolate BBN-metabolizing bacteria from gut microbiota.

To demonstrate that the oxygen-dependent bacterial BBN metabolism also plays a role in the large intestine, we established two different gnotobiotic mouse models based on our *in vitro* data and one of the BBN-metabolizing bacterial isolate (*E. flexneri*): i) We mono-colonized germ-free mice with *E. flexneri* to demonstrate that the bacteria also converts BBN to BCPN in the intestine (Fig. 4f-g and Table S37-38). ii) We repeated the colonization of germ-free mice with *E. flexneri* in the context of a 3-membered bacterial community to demonstrate that BBN metabolism in the large intestine depends on *E. flexneri* also in the presence of other (anaerobic) bacterial strains (Fig. 4i-j and Table S39-40). Together, these experiments suggest that the oxygen concentration in the large intestine may be sufficiently high to support bacterial oxidation of BBN to BCPN. See also the response below and the responses to reviewer 1 and 2 for further details about the developed gnotobiotic mouse experiments.

- Almost certainly, the identification of *Corynebacterium* and *Staphylococcus* species that can catalyze the conversion of BBN to BCPN aerobically is not biologically relevant. Those organisms are skin colonists and present at very low levels in the gut.

We agree that the *Corynebacterium sp.* and *Staphylococcus sp.* are not typical gut bacteria, but rather associated with skin. To test whether their identification may be an artefact of our aerobic and microaerobic enrichment isolation protocol, we mapped the full-length 16S rRNA DNA sequence of the isolates to the 16S rRNA microbiome data of the respective gut communities, from which they were identified (also see response to comment 7 of reviewer 1 and response to reviewer 2). Whereas other bacterial taxa turned out to be likely artefacts of the isolation protocol, we could reliably map *Corynebacterium sp.* and *Staphylococcus sp.* sequences to the microbiome data (Fig. 3c). Although this suggests that these bacteria are present in the murine gut microbiome at detectable levels, we believe that they are likely originating from skin due to murine grooming activity and rather passing through the gut than colonizing it. Despite the fact that even if just passing through, these bacteria could contribute to BBN metabolism (similar to some probiotics). Nevertheless, we used isolated *Escherichia* species, expected to be part of the human and murine gut microbiome, for the followup experiments and to develop our gnotobiotic models.

We describe this analysis in the main text (line 157 – 162 and 209- 224) and materials and methods (line 421 – 434). Results are displayed in Fig. 3c and 4i-j and raw data compiled in Tables S21-22 and Tables S25 and S33.

- I think I understand what the authors are getting at in Fig 4c - they are trying to show that other communities might be involved. But doesn't the data from Fig 2 suggest that most of the BCPN is in the cecum? I am confused about whether enough BCPN is produced in other body sites to make a meaningful contribution to the cancer-inducing pool (again, if BCPN is really driving tumorigenesis in this model). It seems incomplete that it is left unclear which body site is relevant for conversion.

These experiments were originally performed out of the interest to investigate the metabolic potential of bacterial strains colonizing body sites other than the gut. However, we do realise (also from the comments by reviewer 1) that the addition of these results were rather disruptive of the main storyline and does not contribute to the main conclusions of the manuscript. Therefore, we decided to exclude these data in this revision.

Most importantly, in addition to addressing the points above, I think the authors would need to show definitively, using germ-free mice colonized by defined communities +/- a BBN-to-BCPN converting strain (from Fig 3 or 4), that the presence of that strain alone is sufficient to potentiate BBN-induced tumorigenesis. Otherwise the study seems correlative but not definitive; it would be unclear whether the data from Figs 2-4 explain the mechanism of the initial observation.

We do agree with the reviewer and used germ-free mice to develop gnotobiotic models to directly link the *in vitro* work on gut bacterial BBN metabolism to the *in vivo* observation of gut bacterial contribution to BBN toxicokinetics. We established the following three gnotobiotic mouse models based on the *in vitro* data: i) To translate the *ex vivo* data on BBN metabolism of human gut communities to the *in vivo* situation, we colonized germ-free mice with human faecal samples and repeated the BBN exposure experiment with a BBN and a BBN/ABX group. This experiment demonstrated that the microbiome-dependent changes of BBN toxicokinetics is not specific to the murine microbiome and can also be observed for mice carrying a 'humanized' gut microbiota (Fig. 4b, Extended Data Fig. 4c-f, and Table S28-31). ii) To demonstrate that *E. flexneri*, which we isolated from human stool samples based on its *in vitro* capability to metabolize BBN, also

oxidizes BBN in the intestine, we mono-colonized germ-free animals with this strain prior to BBN treatment. We found that *E. flexneri* is capable to also metabolize BBN in the large intestine, leading to the accumulation of BCPN (Fig. 4f-g and Table S37-38). iii) We assembled a BBN-metabolizing and a non-BBN-metabolizing bacterial community for the colonization of germ-free mice and subsequent demonstration of differences in their *in vivo* BBN metabolism (Fig. 4i-j and Table S39-40). Together, these additional gnotobiotic mouse experiments demonstrated the causative role of BBN-metabolizing bacterial strains and communities identified *in vitro* for *in vivo* BBN metabolism and toxicokinetics.

References

1. Zimmermann, M., Zimmermann-Kogadeeva, M., Wegmann, R. & Goodman, A. L. Separating host and microbiome contributions to drug pharmacokinetics and toxicity. *Science* **363**, eaat9931 (2019).
2. Okada, M. & Ishidate, M. Metabolic Fate of N-n-Butyl-N-(4-hydroxybutyl)-nitrosamine and its Analogues: Selective Induction of Urinary Bladder Tumours in the Rat. *Xenobiotica* **7**, 11–24 (1977).
3. Okada, M. & Suzuki, E. Metabolism of Butyl(4-hydroxybutyl)nitrosamine in rats. *GANN Jpn. J. Cancer Res.* **63**, 391–392 (1972).
4. Hashimoto, Y., Suzuki, E. & Okada, M. Induction of urinary bladder tumors in ACI/N rats by butyl(3-carboxypropyl)nitrosamine, a major urinary metabolite of butyl-84-hydroxybutyl)nitrosamine. *GANN Jpn. J. Cancer Res.* **63**, 637-638_1 (1972).
5. HASHIMOTO, Y., SUZUKI, K. & OKADA, M. INDUCTION OF URINARY BLADDER TUMORS BY INTRAVESICULAR INSTILLATION OF BUTYL(4-HYDROXYBUTYL)NITROSOAMINE AND ITS PRINCIPAL URINARY METABOLITE, BUTYL(3-CARBOXYPROPYL)NITROSOAMINE IN RATS. *GANN Jpn. J. Cancer Res.* **65**, 69–73 (1974).
6. Donaldson, G. P., Lee, S. M. & Mazmanian, S. K. Gut biogeography of the bacterial microbiota. *Nat Rev Microbiol* **14**, 20–32 (2016).
7. Albenberg, L. *et al.* Correlation Between Intraluminal Oxygen Gradient and Radial Partitioning of Intestinal Microbiota. *Gastroenterology* **147**, 1055-1063.e8 (2014).

Reviewer Reports on the First Revision:

Referee #1 (Remarks to the Author):

Congratulations on the exciting study, I appreciate the thorough revision. No further comments at this time.

Referee #2 (Remarks to the Author):

This review belongs to the revised version of the manuscript.

Initial major point: Although the BBN model is commonly used to study urothelial carcinogenesis, it represents a highly artificial system that does not represent the regular incorporation of urothelial carcinogens. For many chemical carcinogenesis sequences, it is now known that there is not a single relevant carcinogen, but rather multiple chemical compounds contributing to carcinogenesis (e.g., composition of cigarette smoke).

Therefore, consideration should be given to validating the present results on a more realistic murine carcinogenesis model (e.g., cigarette smoke exposure model) that reflects the full range of the major risk factor of urothelial carcinoma (smoking). On the one hand, this would allow to illuminate and validate the actual influence of the microbiome (and indirectly the impact of nitrosamines in cigarette smoke induced carcinogenesis), but on the other hand, it would also significantly increase the translational and clinical relevance of the present studies, and additionally provide interesting results whether the microbiome significantly influences the carcinogenesis sequence even in the case of a polychemical carcinogen (cigarette smoke). The additional cigarette smoke model would further provide additional benefits and interesting aspects: The authors also studied the microbiome of the upper aerodigestive/gastrointestinal tract and hypothesize that flora alterations may have significant influence on carcinogenesis in these niches as well. Under certain conditions, a cigarette smoke model also allows the study of other cigarette smoke induced carcinomas, especially - squamous cell carcinomas of the head and neck including esophageal carcinomas - lung carcinomas. Like urothelial carcinoma, these tumors classically arise via a chemical-toxic metaplasia/dysplasia carcinogenesis sequence. Therefore, a further step could be to investigate to what extent these sequences are also influenced by the microbiome. This would further massively increase the clinical/medical relevance.

Response of Authors:

We do agree with the reviewer in the potential translational implication of the current study. We also agree that the used BBN-induced urothelial cancer model does not fully reflect the multifactorial nature of (cigarette smoke-induced) urothelial carcinogenesis. However, the goal and main strength of the current study, in our opinion, is the mechanistic dissection of gut microbial contribution to the toxicokinetics of carcinogens at the molecular level, rather than modelling the multifactorial effects that smoke plays in tumorigenesis. Therefore, we used the BBN-induced bladder cancer model, which is one of the best established models to study chemical carcinogenesis, as the reviewer has also pointed out. As the carcinogenesis mechanisms (i.e. BBN to BCPN conversion and accumulation of BCPN in urothelial tissue) have been extensively investigated in

previous studies^{2–5}, we could build on this knowledge to directly quantify the impact of gut bacterial metabolism on these tumorigenic processes. In contrast, the large number of smoke-derived chemicals and additional smoke-related factors that lead to tumorigenesis would make it challenging to interpret results of a smoke-induced cancer model and the impact of the gut microbiota. Therefore, we decided not to establish a smoke-induced tumour model. Instead of the cigarette smoke induced cancer model, we tested 3 additional nitrosamine compounds that have previously been established to model chemical induced tumorigenesis in mice. We could demonstrate that the microbiota impacts the toxicokinetics of these additional nitrosamine compounds the same way as it affects BBN toxicokinetics (Fig. 5a-d, and Table S41). Furthermore, we were able to demonstrate microbiota-dependent bladder tumorigenesis for one of the compounds (EHBN). Altogether, the experiments performed with these additional chemicals, together with the developed gnotobiotic mouse models (described above) both corroborate and generalize our findings. Following the reviewer's suggestion, we believe that future studies should include more complex cancer models in order to facilitate the translation to clinics.

Response of Referee:

I totally agree that cigarette smoke makes it very hard to dissect detailed mechanistic effects of specific carcinogens involved in bladder carcinogenesis. Thus, the well performed experiments provided in Figure 5a-d studying further carcinogens (thus generalizing the results) are in line with initially made suggestions, that in my opinion, sufficiently demonstrate that other important carcinogens involved in bladder carcinogenesis exhibit a similar microbiome dependent mode of bladder cancer induction.

Initial major point: The second major point belongs to the issue of the BBN model. Although the authors clearly show that microbial strains from certain body locations (upper aerodigestive tract, lung, gut) are shared within the same human individual and are able to oxidize BBN to BCPN, it is not proven, that the strains which are found in human isolates are able to propagate the same cancerogenic effects as in mice. Thus, the authors should consider to proof the cancerogenic potential of human isolates in an experimental system, e.g. via isolation of specific BBN->BCPN converting strains, decontamination of murine gut microbiome and transplantation of human strains followed by BBN ingestion with subsequent ABX versus no ABX treatment analysis of murine bladders. (and if applied same procedure in tobacco smoke model!). This would significantly strengthen the biological rationale of the present study.

Response of Authors:

We appreciate the reviewer's comment reproducing the phenotype using human-derived strains and agree that it would be of great relevance to demonstrate their in vivo relevance in a mouse model. To this end, we established three different models using germ-free mice that we colonized with either a BBN-metabolizing human microbiota, a single human-derived, BBN-metabolizing bacterial strain, or two synthetic gut bacterial communities (one with BBN-metabolizing capacity and the other without). These experiments are described in details in the response to comment 4 of reviewer 1 and key results are shown in Fig. 4b and Fig. 4f-j, Extended Data Fig. 4c-f, and Table S28-31 and Table S37-40. Together, these experiments directly link the in vitro work on gut bacterial BBN metabolism and the identified bacterial BBN-metabolizing bacterial isolates to the in vivo observation of gut bacterial contribution to BBN toxicokinetics. Additionally, the established

gnotobiotic mouse models provide an excellent system for future studies aiming at investigating the impact of the gut microbiota on carcinogen metabolism.

Response of Referee: I really appreciate the huge efforts of the authors. Indeed I am satisfied by the new comprehensive results clearly showing that also human strain isolates demonstrate the same carcinogenic capacity in "humanized" mice.

Original minor point: Figure 1B: Dysplasia and CIS panels are not showing convincing cellular atypia. The dysplasia panel is rather showing a squamous metaplasia than a true urothelial dysplasia. It is known that BBN treated mice often develop squamous metaplasia. The image shown as CIS still shows a preserved luminal and basal polarity and there are no remarkable high grade atypia qualifying for CIS. From a pathological point of view there is no need to distinguish between early invasion and invasion. However, the panel with early invasion is not showing a convincing stroma invasion. Thus panels have to be updated with representative images showing dysplasia and CIS. Please remove the category of early invasion. Histopathological analysis could be further strengthened by objective correlates of neoplastic behavior, such as detection of specific oncogenic mutations (TP53 mutations, etc.).

Author reply: We thank the reviewer proposing an improved tumour classification. We followed the reviewer's suggestion and removed the category of 'dysplasia', which is now part of the 'non-cancerous' group. With respect to the reviewer's comment on 'carcinoma in situ (CIS)', we would like to draw their attention on the epithelial changes that are associated with invasive tumours (please check the panel below). We have used these epithelial changes as standard for CIS classification in mice, even though these epithelial changes moderately differ from human CIS. Furthermore, we removed the 'early invasion category' from the tissue classification, because previously categorized 'early invasive' samples are now reclassified as CIS, and because they are not showing convincing stromal invasion, as pointed out by the reviewer. Importantly, our overall pathological findings are not affected by this re-classification and the difference between groups is still highly significant (Fig. 1c, p-value = 1.6E-05).

Response of Referee: I agree with the made changes.

In summary the referee is satisfied with the made changes and believes that the manuscript is substantially improved. Thus publication of the present highly relevant results that are now more generalized is fully supported.

Referee #3 (Remarks to the Author):

Overall I think the authors did a nice job answering my questions and those of the other referees. I am glad they added the germ-free mouse experiments, including the 3-strain community, but I was hoping for these experiments not just to provide evidence showing that *E. flexneri* can metabolize BBN, but that the presence of this strain is sufficient to potentiate BBN-induced tumorigenesis.